# Forging Time Series with Language: A Large Language Model Approach to Synthetic Data Generation

**Cécile Rousseau**
IBM Research Europe
rousseau.cecile@ibm.com

**Tobia Boschi**
IBM Research Europe
tobia.boschi@ibm.com

**Giandomenico Cornacchia**
IBM Research Europe
Giandomenico.Cornacchia1@ibm.com

**Dhaval Salwala**
IBM Research Europe
dhaval.vinodbhai.salwala@ibm.com

**Alessandra Pascale**
IBM Research Europe
apascale@ie.ibm.com

**Juan Bernabe Moreno**
IBM Research Europe
juan.bernabe-moreno@ibm.com

## Abstract

SDForger is a flexible and efficient framework for generating high-quality multivariate time series using LLMs. Leveraging a compact data representation, SDForger provides synthetic time series generation from a few samples and low-computation fine-tuning of any autoregressive LLM. Specifically, the framework transforms univariate and multivariate signals into tabular embeddings, which are then encoded into text and used to fine-tune the LLM. At inference, new textual embeddings are sampled and decoded into synthetic time series that retain the original data's statistical properties and temporal dynamics. Across a diverse range of datasets, SDForger outperforms existing generative models in many scenarios, both in similarity-based evaluations and downstream forecasting tasks. By enabling textual conditioning in the generation process, SDForger paves the way for multimodal modeling and the streamlined integration of time series with textual information. The model is open-sourced at `https://github.com/IBM/fms-dgt/tree/main/fms_dgt/public/databuilders/time_series`.

## 1 Introduction

In the era of foundation models, integrating time-series analysis with language models (LLMs) has emerged as a key research priority, spurring work on specialized models for forecasting, anomaly detection and root cause analysis (Liang et al., 2024). These efforts aim to harness the representational power of LLMs to tackle complex temporal dependencies. Despite these advancements, foundation models for time series still struggle with concept drift, high variability, and face performance degradation over long horizons. In addition, the scarcity of large and diverse time-series datasets often limits model generalization, especially in domains where data collection is expensive or operationally constrained, such as climate science, finance, or plasma physics (Kit et al., 2024).

In this context, *synthetic data generation has emerged as a complementary research direction to improve scalability and model performance, particularly in low-quality or data-scarce settings.* In particular, synthetic data can be leveraged to fine-tune machine learning models on domain-specific distributions, reducing the need to train models from scratch and improving sample efficiency. Indeed,

39th Conference on Neural Information Processing Systems (NeurIPS 2025).

several generative models for time series have been proposed, including VAE-based architectures (Desai et al., 2021; Lee et al., 2023), GANs (Pei et al., 2021; Kidger et al., 2021), and diffusion-based approaches (Zhou et al., 2023). While these methods have shown promise, they typically lack pre-training, require task-specific retraining, and often struggle with long-term dependencies, multivariate coupling, and distributional shifts (Ang et al., 2023).

Recent studies have instead explored the use of LLMs for synthetic tabular data generation (Padhi et al., 2021; Borisov et al., 2022), demonstrating that language models trained on text-encoded data can capture statistical relationships and feature dependencies effectively. However, extending these methods to time series is non-trivial: long temporal windows increase inference and training costs, while temporal and multivariate correlations require more structured modeling.

These limitations highlight the need for novel methodologies that adapt LLMs for time-series generation while addressing temporal, structural, and computational challenges. We introduce *SDForger* (Synthetic Data Forger), a novel framework for generating high-quality *univariate and multivariate time series*, even in data-scarce settings. SDForger leverages foundation models with minimal fine-tuning by operating over compact tabular embeddings derived from functional decompositions.

**Key features and advantages of SDForger:**

- **Compact basis representation** SDForger uses FastICA or PCA to embed time series into low-dimensional tabular data. These capture key temporal and inter-variable structures while decoupling the embedding from sequence length, enabling efficient processing of long signals.
- **Text-to-sequence generation via LLMs** The embedding tables are converted into structured textual prompts and used to fine-tune a language model. A guided inference approach is then used to generate structured embeddings, ensuring that the synthetic data retains the original dataset's statistical properties and feature relationships.
- **Flexible, lightweight architecture** The framework leverages autoregressive LLMs, including lightweight models, and requires only a small number of training instances. Its modular design enables easy adaptation to different generation tasks and architectures, while its compact embedding space ensures fast inference, even for long time-series windows.
- **Multivariate and multimodal readiness** SDForger can model complex multivariate dynamics and supports future extensions to textual conditioning, enabling generation guided by both time-series structure and external language-based context.

By combining structured embeddings with LLM-based generation, SDForger establishes a new paradigm for scalable, interpretable, and high-quality synthetic time-series generation.

Our simulations demonstrate that SDForger not only generates statistically realistic time series but also improves downstream model performance, often matching or exceeding results obtained from real data alone. This is particularly valuable in practical scenarios where access to high-quality data is limited or where distribution shifts make original training data less effective. Compared to state-of-the-art generative models, SDForger achieves competitive or superior performance across a wide range of similarity metrics and utility-based evaluations. Notably, our experiments show that even lightweight, pretrained LLMs (e.g., GPT-2) are sufficient to produce high-quality synthetic data with minimal fine-tuning, highlighting the accessibility, efficiency, and flexibility of our approach.

In the remainder of this paper, we first review related work (Section 2), then detail the SDForger framework (Section 3), describe our evaluation setup (Section 4), present extensive experimental results (Section 5), highlight the flexibility of language models (Section 6), and conclude with key takeaways and future directions (Section 7).

## 2  Related work

**Time-series generation**   Recent advances in time-series generation have introduced a variety of deep generative models, including GAN-based approaches like TimeGAN (Smith and Smith, 2020), state-space models, and vector-quantized architectures such as TimeVQVAE (Lee et al., 2023). While these methods generate realistic sequences, they often struggle with multivariate dependencies, and require training from scratch for each dataset. More recent framework integrate randomly-weighted combinations of time series to improve their pretraining pipeline, e.g. Chronos (Ansari et al., 2024) adapting the Mixup (Zhou et al., 2023) methodology for time series. To address evaluation challenges, TSGBench (Ang et al., 2023) proposes a unified benchmark of similarity, fidelity, and utility metrics.

**Foundation models and LLMs for time series**    Recent advances have introduced foundation models specifically designed for time-series tasks, offering unified frameworks for forecasting, classification, and anomaly detection (Liang et al., 2024). Examples include TFT (Lim et al., 2021), TimeGPT (Garza and Mergenthaler-Canseco, 2023), and Chronos (Ansari et al., 2024), while lightweight models like TTMs (Ekambaram et al., 2024) focus on efficient multivariate forecasting. Parallel efforts have explored adapting LLMs to time-series data by encoding numerical sequences as text (Gruver et al., 2024; Zhou et al., 2023; Jin et al., 2023), enabling zero-shot inference and transfer learning. Notably, LLMTime (Gruver et al., 2024) and GPT4TS (Zhou et al., 2023) retain most of the LLM architecture while fine-tuning only shallow layers, and Time-LLM (Jin et al., 2023) employs reprogramming to adapt to temporal tasks. Despite these innovations, most existing approaches either require task-specific pretraining or struggle to model complex structures and may benefit from synthetic data for improving generalization and robustness.

**Specialized architectures for time-series generation**    Several architectures have been specifically designed to capture temporal and multivariate dependencies in time series. Variational autoencoders such as TimeVAE (Desai et al., 2021) and TimeVQVAE (Lee et al., 2023) use recurrent or vector-quantized structures to model sequential dynamics. GAN-based approaches, including RTSGAN (Pei et al., 2021), SDEGAN (Kidger et al., 2021), and COSCI-GAN (Seyfi et al., 2022), employ recurrent or component-wise disentangled generators to capture complex temporal patterns adversarially. Diffusion-based models, like LS4 (Zhou et al., 2023), generate sequences through learned reverse-time processes. These specialized architectures complement general-purpose time-series generation methods and provide valuable baselines for evaluating synthetic data. However, these architectures are trained from scratch and cannot leverage existing pre-trained language or foundation models, limiting their scalability and adaptability across domains.

## 3  Methodology

In this section, we present our methodology, illustrated in Figure 1. SDForger is divided into three macro-steps: (i) *Preprocessing and embedding* transform the time series into tabular data (i.e., steps 1 to 3 highlighted in purple); (ii) *Fine-tuning and Generation* fine-tune a pre-trained LLM and generate new embedding instances (i.e., step 4 highlighted in green); (iii) *Decoding* reconstruct the original time-series space from the generated embeddings (i.e., steps 5 and 6 highlighted in light blue).

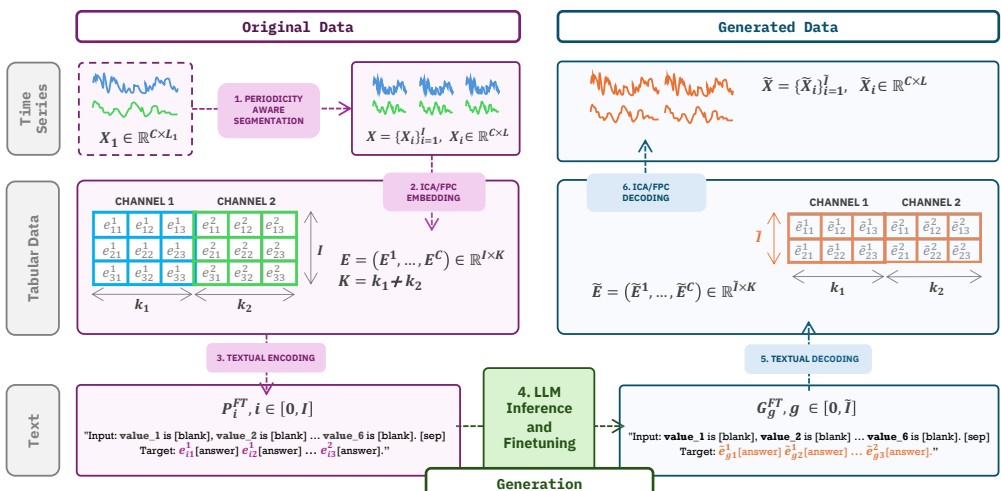

**Figure 1: SDForger pipeline.** Overview of the SDForger generation process. The example illustrates a setting with $I = 3$ input segments, $C = 2$ channels, $k_1 = k_2 = 3$ components, and $\tilde{I} = 2$ generated samples. The model performs periodicity-aware segmentation, extracts embeddings, and embed them into text. An LLM is then fine-tuned to generate embedding sequences, which are finally decoded to reconstruct synthetic time series.

**Notation**    Hereinafter, we introduce some basic notation. Let $X = \{X_i\}_{i=1}^{I}$, $X_i \in \mathbb{R}^{C \times L}$ represent a collection of $I$ instances of a multivariate time series, where each instance has length $L$ and consists

of $C$ channels. The task of *synthetic time-series generation* can now be formally defined as producing $\tilde{I}$ instances of a multivariate time series $\{\tilde{X}_i\}_{i=1}^{\tilde{I}}, \tilde{X}_i \in \mathbb{R}^{C \times L}$ conditioned on the given context $X$. Throughout the paper, we denote by $x_i^c \in \mathbb{R}^L$ the $i$-th instance of channel $c$, and by $X^c \in \mathbb{R}^{I \times L}$ the matrix collecting all instances associated with channel $c$.

**Periodicity-aware segmentation** In cases where only a single time series instance $X_1 \in \mathbb{R}^{C \times L_1}$ is available, we apply a segmentation strategy to artificially create multiple instances. The segmentation procedure facilitates the estimation of the time-series distribution. Specifically, we extract $I$ periodicity-aware windows of fixed length $L < L_1$, aligning cuts with natural cycles and minimizing overlap to enhance independence and diversity. This pre-processing (described in Appendix A.1) transforms the data from a single sequence $X_1$ into a set $X = \{X_i\}_{i=1}^{I}$, where $X_i \in \mathbb{R}^{C \times L}$, preparing the data for the generation task.

### 3.1 From time series to tabular data

To enable tabular generation and analysis, SDForger transforms time series into structured tabular data using *basis decomposition techniques*. Each row represents a time series embedding obtained by projecting the signal onto a set of learned basis functions. Specifically, we adopt two decomposition methods: Functional Principal Components (FPC) (Ramsay and Silverman, 2005) and Fast Independent Component Analysis (FastICA) (Hyvarinen, 1999):

- **FPC** identifies principal modes of variation by performing eigen-decomposition of the covariance operator. It captures directions of maximal variance, preserving correlation across components, showing effectiveness in modeling multivariate longitudinal data (Boschi et al., 2024).
- **FastICA** extracts statistically independent components by maximizing non-Gaussianity. It decomposes a contrast function, uncovering independent latent factors that may not align with the directions of maximal variance.

Formally, for each channel $c$, we assume the instances $(X_1^c, \dots, X_I^c)$ are realizations of continuous functions defined over $\mathcal{T} = [0, L]$. We approximate each $X_i^c$ as a linear combination of $k_c$ basis functions $(b_1^c, \dots, b_{k_c}^c)$, where the choice of basis depends on the decomposition method (non-Gaussianity-based for FastICA, covariance-based for FPC). The embedding coefficients are:

$$e_{ij}^c = \langle X_i^c, b_j^c \rangle_{\mathbb{L}^2} = \int_{\mathcal{T}} X_i^c(t) b_j^c(t) \, \mathrm{d}t,$$

We define the embedding matrix for channel $c$, $E^c \in \mathbb{R}^{I \times k_c}$. By concatenating the embeddings across all channels, we obtain the final embedding table: $E = (E^1, \dots, E^C) \in \mathbb{R}^{I \times K}$ with $K = \sum_{c=1}^{C} k_c$.

Throughout this paper, we refer to the columns of $E$ as embedding features. We denote the $i$-th row of $E$ as $E_i$, corresponding to the embedding vector of instance $X_i$, and the value in its $k$-th column as $e_{ik}$. More details on the choice of $k_c$ are given in Appendix A.2 and Appendix Table D.1.

Notably, both methods offer the advantage that their computational cost depends on the number of instances $I$ and the number of components $k_c$, but not on the instance length $L$. This decoupling allows our algorithm to handle very long time windows without a corresponding increase in computational complexity, ensuring great flexibility and scalability.

### 3.2 Generation of tabular data

Our data generation block consists of three key stages: encoding tabular data into text, fine-tuning an LLM, and generating synthetic embeddings.

#### 3.2.1 From embeddings table to text

LLMs are designed to process textual information. Therefore, applying an LLM to tabular data requires converting each row into a textual format that can serve as a prompt during the fine-tuning stage. Inspired by Donahue et al. (2020), we introduce a *Textual Encoder* responsible for converting tabular instances $E_i$ into structured text representations using a Fill-In-The-Middle template.

**Definition 1 (Textual encoder)** *Let $\mathcal{P}^{\mathrm{FT}} = \{\mathcal{P}_i^{\mathrm{FT}}\}_{i=1}^{I}$ denote the set of fine-tuning prompts, where:*

$$\mathcal{P}_i^{\mathrm{FT}} = \text{``Input: } \bigcirc_{k=1}^K \text{ (value\_}\pi(k) \text{ is [blank],) [sep]}$$
$$\text{Target: } \bigcirc_{k=1}^K \text{ (}e_{i\pi(k)} \text{ [answer])''}$$

*Here, the operator $\bigcirc$ denotes the concatenation and $\pi$ is a random permutation of $K$ elements.*

*Random Feature Order Permutation.* Encoding tabular data into text can introduce unintended positional biases, as LLMs inherently process tokens in sequence. To enforce order independence (Borisov et al., 2022), we apply a random permutation $\pi$ to the encoded feature-value pairs within each instance. This shuffling ensures that the model does not infer any spurious relationships based on the ordering of features within the textual representation. For $K = 2$, an admissible finetuning prompt for $E_i$ is: *"Input: value_2 is [blank], value_1 is [blank] [sep] Target: $e_{i2}$ [answer] $e_{i1}$ [answer]"*

### 3.2.2 Large language model finetuning and inference

**Fine-tuning** By training an LLM on structured text representations of the embedding tables, we enable it to learn meaningful patterns present in the data. Since the optimal number of fine-tuning epochs depends on the number of instances, the embedding dimension, and the LLM architecture, we implement an *early stopping criterion* to prevent overfitting.

**Inference** After fine-tuning, inference is performed by prompting the LLM with structured textual templates that mirror the training format, allowing it to autonomously generate new embedding rows.

**Definition 2 (Textual inference)** *Given the embedding table $E \in \mathbb{R}^{I \times K}$, we define the set of inference prompts at each generation step as $\mathcal{P}^{\mathrm{INF}} = \{\mathcal{P}_g^{\mathrm{INF}}\}_{g=1}^G$ where:*

$$\mathcal{P}_g^{\mathrm{INF}} = \text{``Input:} \bigcirc_{k=1}^K \text{ ( value\_}\pi(k) \text{ is [blank], )  [sep] Target:''}$$

We use a multinomial distribution sampling strategy to reduce repetition and generate more creative and diverse outputs. The model draws from its learned token probability distribution at each step, guided by the temperature parameter, which controls sampling variability. As a result, all values are internally generated by the LLM in a fully conditional and self-contained manner, highlighting the model capacity to internalize statistical and structural patterns from compact embeddings and synthesize coherent time series without external noise injection or sampling routines.

At each inference step, we generate a batch of $G$ synthetic instances, repeating the process until the desired number of sequences is obtained or a stopping criterion is met. We denote the set of all generated text instances as: $\mathcal{G} = \{\mathcal{G}_1, \ldots, \mathcal{G}_G\}$. Ideally, the fine-tuned LLM should generate text instances in the following format: $\mathcal{G}_g = \bigcirc \left( \mathcal{P}_g^{\mathrm{INF}}, \bigcirc_{k=1}^K (\tilde{a}_{g\pi(k)} \text{ [answer]} ) \right)$ where $\pi$ is the random permutation used in $\mathcal{P}_g^{\mathrm{INF}}$, and $\{\tilde{a}_{g\pi(k)}\}_{k=1}^K$ are the $K$ numerical values inferred, which form the generated embedding table.

**Retrieve embedding from text** Given a generated text instance $\mathcal{G}_g \in \mathcal{G}$, we reconstruct the corresponding tabular data by mapping the inferred embedding values $\{\tilde{e}_{g\pi(k)}\}_{k=1}^K$ to their respective features. Each textual entry is split into feature-value pairs using *"[answer]"* as a delimiter. Missing or unrecognized features are assigned the placeholder `"NaN"`. For a specific channel $c$, the output of an inference step $s$ is the reconstructed embedding matrix: $\tilde{E}^{c,s} \in \mathbb{R}^{G \times k_c}$, where each row corresponds to a generated instance $\mathcal{G}_g$ and each column represents an inferred embedding feature associated with channel $c$. To track all generated embeddings up to step $s$, we define: $\tilde{E}^{c,\leq s}$.

**In-generation filtering and stopping criterion** At each inference step $s$, we apply an online filtering procedure that validates generated embeddings without requiring reconstruction into the time-series domain, ensuring efficient real-time evaluation. Specifically, the reconstructed embedding matrices $\left(\tilde{E}^{1,s}, \ldots, \tilde{E}^{C,s}\right)$ are filtered based on three criteria: 1) Instances with missing values are discarded, as they prevent accurate reconstruction; 2) Duplicated instances are discarded to maintain diversity in the generated dataset; 3) Significantly diverging instances are discarded. This combined filtering procedure not only enforces diversity and validity among the generated instances but also provides a diagnostic signal: if a substantial fraction of samples is rejected, it may indicate that the fine-tuned LLM requires further training or more representative data. Representative examples of

discarded instances and details on the divergence detection procedure are provided in Appendix A.3, while Appendix Table D.2 reports the rejection rates observed in a representative generation scenario, illustrating the balance between filtering rigor and sample diversity.

In generation mode, SDForger employs a dynamic stopping criterion that continues generating batches of $G$ text instances as long as sufficient diversity is preserved among the generated samples (Appendix A.4). However, for consistent comparison with baseline methods across all simulation scenarios, we fix the number of generated instances $\tilde{I}$ across all algorithms. If we denote by $S$ the final inference step, then the output of the generation process is the complete embedding table $\tilde{E} \in \mathbb{R}^{\tilde{I} \times K}$, where, for each channel $c$, $\tilde{E}^c = \tilde{E}^{c, \leq S}$.

### 3.3 Decoding: from tabular embeddings to time series

Given $\tilde{E}$, the time-series representation of generated embeddings can be efficiently recovered due to the reversible nature of the embedding technique used. For the channel $c$, given the generated coefficients $\tilde{e}_{ij}^c$ and the corresponding basis system $(b_1^c, \ldots, b_{k_c}^c)$, the reconstructed time series are computed as follows Kokoszka and Reimherr (2017): $\tilde{x}_i^c = \sum_{j=1}^{k_c} \tilde{e}_{ij}^c b_j^c$.

This formulation ensures that each generated embedding is decoded back to the original space, resulting in $\tilde{I}$ synthetic instances of a multivariate time series $\tilde{X}_i \in \mathbb{R}^{C \times L}$.

## 4 Evaluation methodology

**Evaluation metrics**  Evaluating synthetic time-series data requires balancing *realism*, *usability*, and *efficiency*. A strong generative model should replicate key properties of real data while supporting downstream tasks such as forecasting. We adopt a comprehensive evaluation framework comprising two categories: *similarity metrics* and *utility metrics*.

- **Similarity metrics**, inspired by Ang et al. (2023), assess how closely the generated data matches the real data in terms of distribution, structure, and behavior. They fall into two subtypes: (i) **Feature-based metrics** which include *Marginal Distribution Difference (MDD)*, *Auto-Correlation Difference (ACD)*, *Skewness Difference (SD)*, and *Kurtosis Difference (KD)*, assess how well synthetic data retains key statistical properties of real data; (ii) **Distance-based metrics** include *Euclidean Distance (ED)*, *Dynamic Time Warping (DTW)*, and *SHAP-RE* (SHR), a shapelet-based reconstruction error. They quantify the similarity between synthetic and real data in raw feature space or temporal alignment. Formal definitions are provided in Appendix B.
- Utility metrics assess the effectiveness of synthetic data in downstream tasks. Specifically, we fine-tune Tiny Time Mixers (TTM) (Ekambaram et al., 2024), a recent foundation model for multivariate time series, under four settings: (1) zero-shot (no fine-tuning) (2) real data only, (3) synthetic data only, and (4) real data augmented with synthetic data. This setup quantifies the impact of synthetic data on model transferability, data efficiency, and robustness.

**Evaluation protocols**  We consider three distinct evaluation settings to assess the generative capabilities of SDForger across different structural assumptions:

- **Multisample generation** aims to produce new instances by combining patterns from multiple existing time series. This setting reflects scenarios such as generating experimental samples, weather profiles, or patient trajectories from heterogeneous observations. It emphasizes diversity and generalization in data-rich contexts.
- **Univariate generation** focuses on learning from a single time series to generate plausible alternative versions. This is useful for simulating counterfactual histories, seasonal variations, or stress-test scenarios in domains like finance, weather, and demand forecasting.
- **Multivariate generation** evaluates the ability to jointly generate multiple interdependent channels. It reflects real-world settings, such as energy systems, traffic flows, or sensor networks, where channel interactions and cross-correlations are crucial for realism and downstream utility.

In the multisample case, multiple instances are available by design. In contrast, for univariate and multivariate settings, only one instance is provided; therefore, we first apply the period-aware segmentation procedure described in Section 3 to extract multiple windows from each channel.

**Parameter settings**   We summarize here the hyperparameters for SDForger. We fix the embedding dimension to $k = 3$ for the *multisample* and *univariate setting*. The LLM used for generation is GPT-2 [1], fine-tuned with Adam (Diederik, 2014) optimization, a learning rate of $8 \times 10^{-5}$, batch size 32, and a maximum of 200 epochs. Early stopping criteria is applied based on the best validation loss computed every 5 steps, patience set to 5, randomly choosing $20\%$ of the data as a validation set.

**Baselines**   We evaluated SDForger's performance against several baseline models for synthetic time series generation, covering different approaches. *Variational autoencoders*: TimeVAE (Desai et al., 2021), which models temporal dependencies with a recurrent VAE architecture, and TimeVQVAE (Lee et al., 2023), which incorporates vector quantization for better capturing discrete temporal patterns; *generative adversarial networks*: RTSGAN (Pei et al., 2021), which uses recurrent components for adversarial training, and SDEGAN (Kidger et al., 2021), which models time series as solutions to stochastic differential equations; and a *diffusion-based model*: LS4 (Zhou et al., 2023), which generates sequences via a learned reverse-time diffusion process. Hyperparameters for all baseline competitors follow those reported in their original papers, except for SdeGAN, for which we fix the number of training iterations to 1000 to balance convergence and computational cost.

**Datasets**   We evaluated SDForger models using 12 publicly available datasets from various domains, including energy, transport, industry, weather, and finance, with sampling frequencies ranging from 2 minutes to monthly. The datasets, sourced from the Monash Time Series Forecasting Repository and other public domains, include both stationary and non-stationary time series, reflecting diverse temporal dynamics. Detailed information is provided in Appendix C.

# 5   Results

Following, we discuss results on **Similarity-based** (Section 5.1), **Utility-based metrics** (Section 5.2), and a condensed ablation study (Section 5.3). Complete ablations are provided in Appendix D.

## 5.1   Similarity-based metrics results

The similarity-based results aggregated for the multisample and univariate settings are reported in Table 1, with detailed per-dataset scores provided in Appendix Tables D.10, D.11, D.12, D.13, and D.14.

**Overall performance.** Different generative models exhibit complementary strengths: for instance, TimeVAE performs well on distribution-based metrics, while TimeVQVAE excels on distance-based measures such as Euclidean Distance and DTW. In contrast, *SDForger achieves consistently strong and balanced performance across both metric categories*, maintaining high scores without overfitting to either statistical or structural similarity (Table 1). This balanced behaviour is further confirmed by the normalized average scores per metric group and the average rank values. Such consistency indicates that SDForger not only preserves key statistical features but also captures the underlying temporal and distributional structure of the data, demonstrating strong generalization and robustness across heterogeneous temporal domains. By decoupling representation learning from generation, SDForger captures long-range dependencies while maintaining statistical realism, ultimately producing temporally coherent and domain-consistent synthetic samples.

**Robustness to evaluation protocols** Comparing multisample and univariate settings, we observe that model rankings and relative performances remain largely consistent, suggesting that SDForger is robust to variations in the evaluation protocol. This stability is an important advantage in practice, where test-time conditions may vary.

**ICA vs. FPC** The ICA embedding strategy consistently leads, particularly on distance-based metrics. The superior performance of the ICA-based variant likely comes from the nature of the components it produces. Unlike FPC, which orders components by explained variance and often concentrates most information in the first few components, ICA explicitly seeks statistically independent components. This tends to produce a more balanced and disentangled basis decomposition, where each component carries distinct information that have similar importance for data reconstruction. For our LLM-based generation pipeline, this disentanglement appears advantageous because the model can learn a joint

---

[1] https://huggingface.co/openai-community/gpt2

**Table 1: Aggregated performance comparison in the multisample and univariate settings.** Metrics include raw similarity scores and normalized averages (in $[0-1]$) for each metric group, plus the average rank. Lower values are better. **Bold** indicates the best performance per column, and underlined indicates the second-best.

| | | Feature-based | | | | Distance-based | | | Norm. Avg. | | |
|---|---|---|---|---|---|---|---|---|---|---|---|
| | | **MDD** | **ACD** | **SD** | **KD** | **ED** | **DTW** | **SHR** | **Feat.** | **Dist.** | **Rank** |
| MULTISAMPLE | SDF-ICA$_3$ | 0.244 | 1.180 | 0.869 | 2.384 | 16.669 | 12.373 | 6.870 | 0.224 | 0.074 | 3.143 |
| | SDF-FPC$_3$ | 0.255 | 2.166 | 1.323 | 4.299 | 17.749 | 11.921 | 16.537 | 0.562 | 0.100 | 4.714 |
| | TimeVAE | **0.227** | **0.259** | **0.507** | **1.697** | 18.041 | 11.625 | 14.021 | **0.000** | 0.094 | **2.143** |
| | TimeVQVAE | 0.371 | 5.466 | 1.327 | 3.889 | **13.661** | 10.167 | **2.030** | 0.873 | **0.000** | 3.714 |
| | RtsGAN | 0.279 | 1.769 | 0.612 | 2.300 | 16.084 | 11.859 | 5.631 | 0.231 | 0.058 | 2.857 |
| | SdeGAN | 0.240 | 2.098 | 1.404 | 4.091 | 37.174 | 33.391 | 51.678 | 0.540 | 0.693 | 5.286 |
| | LS4 | 0.276 | 6.150 | 1.243 | 4.852 | 44.389 | 31.806 | 160.403 | 0.789 | 0.977 | 6.143 |
| UNIVARIATE | SDF-ICA$_3$ | 0.306 | **1.396** | 0.671 | 1.382 | 18.802 | 12.435 | 4.856 | 0.149 | 0.070 | **2.429** |
| | SDF-FPC$_3$ | 0.308 | 1.480 | 0.801 | 1.690 | 19.340 | 12.809 | 5.452 | 0.354 | 0.084 | 4.000 |
| | TimeVAE | 0.288 | 2.013 | **0.611** | **1.245** | 20.778 | 12.126 | 18.534 | **0.066** | 0.158 | 2.714 |
| | TimeVQVAE | 0.433 | 4.330 | 0.740 | 2.052 | **15.438** | 11.250 | **2.217** | 0.707 | **0.000** | 3.571 |
| | RtsGAN | 0.363 | 2.389 | 0.776 | 1.325 | 18.951 | 12.926 | 5.464 | 0.384 | 0.081 | 4.000 |
| | SdeGAN | **0.267** | 3.659 | 0.813 | 1.542 | 42.017 | 38.541 | 65.557 | 0.390 | 0.979 | 5.143 |
| | LS4 | 0.298 | 6.041 | 0.855 | 2.457 | 40.362 | 24.262 | 69.751 | 0.797 | 0.805 | 6.143 |

**Table 2: Utility evaluation via fine-tuned forecasting models.** TTM forecasting performance on downstream tasks using different training sources: zero-shot, original data, generated data, and a combination of original and generated data. Results are reported for 3 multivariate datasets: *bikesharing* (target: `count`, control: `temperature`, `humidity`), *etth1* (target: `HUFL`, control: `MUFL`, `OT`), and *traffic* (target: `junction1`, control: `junction2`, `junction3`). Metrics include RMSE, MASE, WQL, and average rank (lower is better). **Bold** highlights the best result within each row group; underlined the second best; **bold+underlined** the overall best.

| | | bikesharing | | | etth1 | | | traffic | | | |
|---|---|---|---|---|---|---|---|---|---|---|---|
| | | **RMSE** | **MASE** | **WQL** | **RMSE** | **MASE** | **WQL** | **RMSE** | **MASE** | **WQL** | **Avg. Rank** |
| | 0-shot | 0.728 | 2.150 | 0.287 | 0.678 | 2.132 | 0.255 | 0.708 | 1.555 | 0.255 | 1.78 |
| | Original Data (OD) | **0.495** | **0.822** | **0.178** | **0.658** | **1.820** | **0.232** | **0.702** | 1.995 | 0.283 | 1.22 |
| GENERATED | SDF-ICA | **0.514** | 0.899 | 0.194 | **0.626** | **1.820** | 0.224 | 0.655 | 1.849 | 0.262 | **2.00** |
| | SDF-FPC | 0.527 | 0.926 | 0.200 | 0.650 | 1.887 | 0.232 | 0.662 | 1.837 | 0.262 | 3.22 |
| | TimeVAE | 0.566 | 0.983 | 0.211 | 0.690 | 2.268 | 0.269 | 0.738 | 2.078 | 0.296 | 5.33 |
| | TimeVQVAE | 0.520 | **0.867** | **0.188** | **0.626** | 1.874 | 0.227 | 0.702 | 1.995 | 0.283 | 2.67 |
| | RtsGAN | 0.710 | 1.261 | 0.275 | 0.770 | 2.271 | 0.291 | **0.597** | **1.574** | 0.225 | 4.67 |
| | SdeGAN | 0.572 | 0.995 | 0.214 | 0.688 | 2.262 | 0.263 | 0.629 | 1.715 | 0.243 | 4.00 |
| | LS4 | 0.839 | 1.468 | 0.318 | 0.642 | 1.977 | 0.236 | 0.917 | 2.595 | 0.369 | 5.89 |
| ORIGINAL + GEN | SDF-ICA + OD | **0.487** | **0.801** | **0.173** | 0.642 | **1.746** | 0.226 | 0.750 | 2.110 | 0.301 | 3.22 |
| | SDF-FPC + OD | 0.493 | 0.829 | 0.179 | 0.666 | 1.754 | 0.231 | 0.743 | 2.087 | 0.297 | 4.78 |
| | TimeVAE + OD | 0.492 | 0.814 | 0.176 | 0.654 | 1.752 | 0.228 | 0.721 | 2.039 | 0.290 | **2.89** |
| | TimeVQVAE + OD | 0.495 | 0.804 | 0.174 | 0.678 | 1.887 | 0.242 | 0.724 | 2.043 | 0.291 | 4.56 |
| | RtsGAN + OD | 0.498 | 0.819 | 0.177 | 0.637 | 1.872 | 0.231 | 0.607 | **1.647** | **0.234** | 3.44 |
| | SdeGAN + OD | 0.495 | 0.837 | 0.181 | **0.620** | 1.843 | **0.224** | **0.605** | 1.716 | 0.242 | 3.33 |
| | LS4 + OD | 0.497 | 0.822 | 0.178 | 0.660 | 1.819 | 0.233 | 0.745 | 2.111 | 0.300 | 5.56 |

distribution over a set of factors that all have the same "power". Thus, the results suggest that the LLM is indeed better equipped to model the joint distribution when presented with independent factors rather than a hierarchy of variance-ordered components.

## 5.2 Utility-based metrics results

Table 2 presents the utility evaluation, where we assess the practical value of synthetic data by fine-tuning TTM on different multivariate training sources. For LS4, we modified the architecture to support multivariate generation. Furthermore, we adjust the embedding dimension $k$ in SDForger according to the complexity of each dataset, setting $k$=3 for *bikesharing*, $k$=7 for *etth1*, and $k$=5

for *traffic*. To determine these values, we conducted a small ablation study to identify the optimal embedding dimension for each dataset (see Table D.7).

SDForger demonstrates strong performance across datasets, as evidenced by its top average rank, with notable results on *bikesharing* and *etth1*. In *bikesharing*, synthetic data from SDForger alone yields competitive scores, and combining it with real data leads to the best overall performance across metrics. On *etth1*, SDForger-generated data surpasses original data in RMSE and WQL, suggesting it captures critical temporal and statistical structure. The hybrid setting (original + generated) maintains this advantage and further improves MASE. Performance on *traffic* is more nuanced. Here, fine-tuning on real data is less effective, and GAN-based methods outperform others. Nevertheless, SDForger remains competitive, especially when using synthetic data alone. This suggests that the test distribution may deviate significantly from the training set, making traditional fine-tuning less useful. Indeed, high-quality synthetic data can act as a valuable supplement or even an alternative.

In no scenario does synthetic data degrade downstream performance, underscoring the reliability and utility of SDForger-generated samples across varied forecasting contexts.

## 5.3 Ablation

**Effect of embedding dimension $k$** Appendix Table D.6 presents an ablation study on the number of components $k$ used in SDForger's embedding space. A compact embedding with $k{=}3$ offers strong performance across both *multisample* and *univariate* settings, indicating that a small number of components is often sufficient to capture core temporal and structural patterns. However, the optimal value of $k$ may vary across datasets, with more intricate dynamics potentially requiring higher-dimensional representations. In practice, users may also opt to select $k$ based on a desired percentage of explained variance, adapting the representation to specific application needs.

**Domain-level insights** Appendix Tables D.3 and D.4 summarize the average normalized similarity scores per dataset. SDForger models achieve strong performance across a wide range of domains. Structured datasets such as *Energy*, *Appliances*, and *Weather* exhibit particularly high scores. In contrast, domains such as *Tourism*, *Traffic*, and *Finance* present greater challenges, likely due to their increased irregularity and noise. Nonetheless, SDForger maintains competitive results even in more complex settings, underscoring the flexibility of the proposed architecture.

**Generation efficiency** Appendix Table D.5 reports the average time to generate univariate sequences for three targets from the *Bikesharing* dataset, across two window lengths. SDForger is substantially faster than all competitors, often by one to two orders of magnitude. TimeVAE is the closest competitor but remains over $4\times$ slower. Notably, unlike GAN-based competitors, SDForger's generation time is independent of sequence length and scales with the number of embedding components ($k$). A minor exception occurs at $k = 3$, where the reduced latent expressivity increases LLM fine-tuning time. Overall, SDForger achieves state-of-the-art efficiency without compromising quality.

**LLM comparison** `GPT-2` (124M) achieves performance on par with, and sometimes better than larger and more recent models such as `granite-3.0` (2B) and `phi-3.5` (3.8B) (Appendix Table D.9). This shows that SDForger's pipeline is effective even leveraging lightweight models. While runtime cost grows with model size, SDForger remains efficient compared to baselines (Appendix Table D.8).

**Filtering Procedure.** Appendix Table D.2 reports rejection statistics for a representative generation scenario. We observe that the overall discard rate remains consistently low ($< 2\%$) across settings, indicating that most generated embeddings fall within a plausible norm range. The proportion of missing values increases with larger embedding dimensions, reflecting a higher likelihood of incomplete generations in longer textual outputs. Notably, the $\ell_2$ norms of accepted samples closely match those of the original embeddings, while discarded ones exhibit markedly higher values, confirming that the filtering procedure effectively removes divergent or anomalous generations.

## 6 Shaping time series with language

SDForger is designed to naturally incorporate textual information, making it well-suited for state-of-the-art time-series generation that embraces additional multi-modal inputs. To explore this, we conduct an experiment using the *bikesharing* dataset. Coming from the intuition that these variables stem from a common physical process and may share latent components, we embed their three

channels (temperature, count, and humidity) into a shared ICA basis. We incorporate the channel information in the textual encoder: ''`Condition: data is temp [sep] Input: value_1 is [blank]...[sep] Target: `$e_{i1}$` [answer]...`''

This conditioning strategy enables SDForger to generate channel-specific sequences with high fidelity. For instance, using a longitudinal $k$-nearest neighbor classifier (Ramos-Carreño et al., 2024) trained on real data, we achieve an accuracy of $0.81$ in identifying the generated curves (see Figure 2). These results highlight SDForger's strong generative capacity and its ability to integrate and respond to textual cues, positioning it as a flexible and powerful baseline for multimodal time-series synthesis.

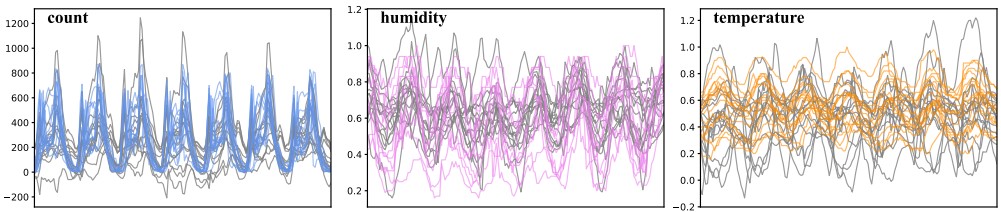

**Figure 2: Text-Conditioned Generation with SDForger.** Visualization of 10 original (grey) and synthetic samples per channel from the *bikesharing* data. Synthetic data is generated using conditional prompts: "`Condition: data is cnt` (blue)", "`Condition: data is hum` (pink)", and "`Condition: data is temp` (orange)".

# 7    Conclusions

We introduced **SDForger**, a flexible and efficient framework for generating synthetic multivariate time series using large language models. By combining compact functional embeddings with textual conditioning, SDForger enables high-quality generation even in data-scarce settings. Extensive evaluations across multiple datasets and tasks demonstrate that SDForger consistently achieves strong similarity scores and enhances downstream forecasting performance—often matching or surpassing results obtained from real data and outperforming state-of-the-art baselines.

Ablation studies confirm the robustness of the framework across embedding strategies, dimensionality choices, and LLM architectures. SDForger is also highly efficient, with significantly lower generation times compared to its competitors. Moreover, by leveraging LLMs, SDForger enables seamless integration with textual prompts, paving the way for multimodal time-series generation, where natural language can guide not only content but also structure, semantics, or temporal context.

We believe SDForger can be further improved. Its modular design is intentionally built to support flexible experimentation, making it easy to explore enhancements or tailor components to specific needs. We see several promising directions:

- **Embedding Strategies** While our current approach relies on linear methods like FastICA and FPC, future work could explore more expressive, nonlinear embeddings (e.g., AE) or multivariate-aware methods like Multivariate FPCA or Multivariate Singular Spectrum Analysis to better capture temporal and inter-channel dependencies.
- **Parameter-Efficient Fine-Tuning** We currently use full fine-tuning for the LLM. However, using too many components relative to the number of instances can lead to unstable fine-tuning and reduced generation quality. Incorporating PEFT techniques such as LoRA or adapters could improve scalability, efficiency, and facilitate domain adaptation.
- **Extension to encoder-only models** Our current implementation supports only autoregressive LLMs; future work would extend the framework to encoder-only models and different generation paradigms such as masked token prediction.
- **Extended Utility Evaluation** While we focus on forecasting, SDForger could be evaluated and optimized for broader downstream tasks such as classification or anomaly detection.
- **Context and Covariate Integration** By design, SDForger supports integration of external covariates (e.g., categorical or textual data). Expanding this functionality could enable richer conditional generation, and multimodal transfer learning (see Section 6).

In summary, SDForger offers a flexible foundation, and we see meaningful opportunities to improve it both architecturally and in terms of task generalization.

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

# Appendix

## A   Implementation details

### A.1   Data preprocessing: segmentation

In many scenarios, each channel consists of a single historical time series, i.e., $I_0 = 1$. However, to estimate embeddings that effectively capture the temporal distribution, multiple instances per channel are necessary. Therefore, when $I_0 = 1$, we segment each channel into multiple overlapping windows. Specifically, for each channel $c$, we construct $I_c$ windows of fixed length $L_c$, where $L_c < L_0$. Without loss of generality, for simplicity in notation, we assume $L_c$ and $I_c$ are identical across all channels and denote them as $L$ and $I$, respectively.

To ensure robust learning of the embedding distribution, $I$ must be sufficiently large. Our experiments indicate that even $I = 15$ (i.e., 15 instances) suffices for this purpose. Once $I$ and $L$ are fixed, we segment the time series while minimizing the overlap between consecutive windows. The overlap step is determined by the dominant periodicity $P$ of the channel, ensuring that window transitions align with intrinsic temporal cycles.

The set of extracted windows $\mathcal{W}$ is formally defined as:

$$\mathcal{W} = \{X[t : t + L] \mid t = 0, s, 2s, \dots, L_0 - L\} \tag{1}$$

where the step size $s$ is computed as: $s = \max(1, \lfloor \frac{L_0 - L}{I - 1} \rfloor)$ and then adjusted to be the nearest multiple of $P$ to maintain consistency in periodic structure.

To determine the dominant periodicity $P$, we employ the Autocorrelation Function (ACF), which quantifies the similarity between the time series and its lagged versions at different time shifts. This method is robust to noise and remains effective even when periodicity is not strictly stationary.

The estimation of $P$ follows these steps:

1. Compute the ACF and identify significant peaks, excluding lag $0$.
2. Rank the detected peaks by their autocorrelation values.
3. Select the highest-ranked period $P$ such that $P < L/2$.

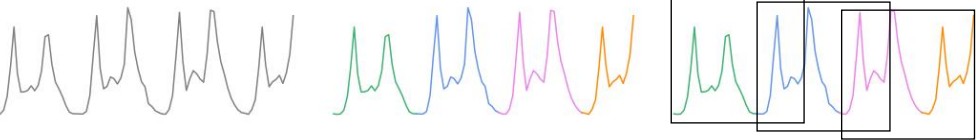

**Figure A.1: Periodicity-aware segmentation.**

By leveraging this periodicity-aware segmentation strategy, we ensure that the extracted windows align with the natural cycles of the time series. Moreover, this approach minimizes window overlap, maximizing their independence and diversity and facilitating more effective embedding computations for downstream generative tasks.

This pre-processing transforms the data from a single sequence $X_1$ into a set $X = \{X_i\}_{i=1}^{I}$, where each $X_i \in \mathbb{R}^{C \times L}$, preparing the data for the generation task.

### A.2   Choice of the number of components

The choice of $k_c$ determines how well the basis representation approximates the original time-series channel $c$. Our framework allows the user to either select the smallest $k_c$ that explains a predefined percentage of the total variance of the original time series or manually specify $k_c$.

There exists an inherent trade-off in selecting $k_c$. A higher $k_c$ captures more of the total variability but can hinder the LLM's ability to model the underlying distribution during the generation phase. Moreover, if $k_c$ is too large, the generated samples may become overly similar to the original data,

limiting diversity and the introduction of novel patterns. Conversely, choosing $k_c$ too small risks omitting essential structures and temporal characteristics, degrading the reconstruction quality.

It is important to note that the nature of the components differs between FPC and FastICA. FPC forms a parsimonious basis system, where a few components typically suffice to capture most of the variability, with components ordered by the amount of variance they explain—early components being systematically more informative. In contrast, FastICA components are unordered: each component contributes independently, without a hierarchical importance structure. As a result, FastICA generally requires more components to achieve a similar reconstruction quality compared to FPC. However, this property also makes FastICA embeddings more robust during generation, as information is distributed more evenly across components, reducing the risk that a few badly generated components disproportionately affect the synthesized curves.

To provide a quantitative intuition, Appendix Table D.1 reports the proportion of variance retained across embedding dimensions for both decomposition methods. This analysis highlights how the variance explained increases with $k$, and how FPC typically achieves higher cumulative variance with fewer components, while FastICA distributes information more evenly across dimensions.

In practice, we recommend keeping the total number of components $K$ reasonably small, particularly when the number of training instances is limited. Empirically, with a training set of 30 instances, setting $K > 25$ often results in unstable fine-tuning and an increased rate of discarded samples due to low-quality generation. This limitation stems from the LLM's reduced ability to model high-dimensional embeddings under data-scarce conditions effectively.

### A.3  In-generation filtering

**Missing values and duplicated instances.**    To illustrate the filtering logic, we report three concrete examples of generated prompts with embedding dimension $K = 4$. Following the inference template:

$$\mathcal{P}_g^{\mathrm{INF}} = \text{``Input: } \bigcirc_{k=1}^K \text{ ( value\_}\pi(k) \text{ is [blank], ) [sep] Target:''},$$

the model produces the following textual generations:

- **Prompt 1 (valid)**
  ```
  Input:  value_2 is [blank], value_4 is [blank], value_1 is [blank],
  value_3 is [blank] [sep] Target:  0.125 [answer] -0.084 [answer] 0.217
  [answer] 0.041 [answer]
  ```

- **Prompt 2 (duplicated)**
  ```
  Input:  value_4 is [blank], value_1 is [blank], value_2 is [blank],
  value_3 is [blank] [sep] Target:  -0.084 [answer] 0.217 [answer] 0.125
  [answer] 0.041 [answer]
  ```

- **Prompt 3 (missing value)**
  ```
  Input:  value_1 is [blank], value_3 is [blank], value_2 is [blank],
  value_4 is [blank] [sep] Target:  0.182 [answer] 0.095 [answer] -0.012
  [answer]
  ```

In this example, the filtering stage identifies *Prompt 2* as a duplicate of *Prompt 1* and discards it, while *Prompt 3* is removed because it does not contain all the targets' coefficients. Only *Prompt 1* is retained for reconstruction. This simple yet effective procedure ensures that the generated embedding tables remain diverse and valid before decoding into the time-series domain.

**Diverging instances**    To ensure the quality of synthetic data, we discard generated instances whose embedding coefficients significantly deviate from the distribution of the original data. Specifically, we compute the squared $\ell_2$-norm of each embedding vector and compare it to the norms of the original embeddings. This criterion efficiently filters out extreme outliers in the latent space, without requiring reconstruction into the time-series domain.

Formally, for each channel $c$, let $\hat{E}^{c,s} = E^c \cup \tilde{E}^{c,\leq s-1}$ be the matrix containing both original embeddings and all previously accepted generated embeddings up to inference step $s - 1$, with $\hat{E}^{c,0} = E^c$. Denote by $N^{c,\mathrm{old}}$ and $N^{c,\mathrm{new}}$ the sets of squared Euclidean norms of the rows of $\hat{E}^{c,s-1}$

and the newly generated matrix $\tilde{E}^{c,s}$, respectively. For each newly generated row $i$, we compute its norm and accept it only if:

$$q_1 - 3 \cdot \text{IQR} \leq N_i^{c,\text{new}} \leq q_3 + 3 \cdot \text{IQR},$$

where $q_1$ and $q_3$ are the first and third quartiles of $N^{c,\text{old}}$, and $\text{IQR} = q_3 - q_1$. An instance is retained only if this condition is satisfied across all channels $c$.

This norm-based strategy is particularly well-motivated when using FPCs, due to the orthonormality of the basis. Let $\mathcal{X}_i^c$ be a time series in channel $c$ and $b_j^c$ the corresponding FPC basis. Then the $\mathbb{L}^2$-norm of $\mathcal{X}_i^c$ can be approximated by the Euclidean norm of its FPC coefficients:

$$\int_{\mathcal{T}} (\mathcal{X}_i^c)^2 \, \mathrm{d}t = \|\mathcal{X}_i^c\|_{\mathbb{L}^2}^2 = \sum_{j=1}^{\infty} \langle \mathcal{X}_i^c, b_j^c \rangle_{\mathbb{L}^2}^2 \approx \sum_{j=1}^{k_c} (e_{ij}^c)^2,$$

where $e_{ij}^c$ are the FPC embedding coefficients. This justifies the norm-based filtering as a direct proxy for detecting time-series samples with unusually high or low energy.

While filtering based on coefficient norms does not guarantee full statistical fidelity to the original time-series distribution, it serves as an effective mechanism to remove extreme outliers without reconstruction. Combined with additional checks for missing values and duplicates, this step helps preserve both the *diversity* and *relevance* of the generated data. A high rejection rate may indicate insufficient LLM fine-tuning or poor generalization, suggesting the need for more representative training data or additional training steps.

### A.4 Stopping criterion

The stopping criterion monitors the diversity of the generated norms across all channels to determine when the generation process should stop. When using FPC, these norms correspond to the $\mathbb{L}^2$-norms of the generated curves. When using FastICA, they correspond to the norms of the embedding coefficient vectors; although not directly related to the curve norms, they still provide a useful proxy for identifying over-sampling and loss of variability.

At inference step $s$, for each channel $c$, let $u^c$ denote the number of unique values in $N^{c,\text{old}}$ (the set of accepted norms up to step $s$), rounded to the fourth decimal place. Let $\tilde{I}$ be the total number of valid instances generated so far. We define the *diversity score* for channel $c$ as:

$$D^c = u^c / \tilde{I}.$$

The diversity score provides a quantitative measure of how much variability remains in the generated norms. We track $D^c$ at each inference step, and the stopping condition is triggered when:

$$\max_c D^c < \lambda_{\text{stop}} \quad \text{or} \quad \tilde{I} > \tilde{I}_{\text{max}}.$$

In other words, generation stops either when the maximum diversity score across channels falls below a predefined threshold $\lambda_{\text{stop}}$, indicating reduced novelty, or when the total number of generated instances exceeds a maximum cap $\tilde{I}_{\text{max}}$.

Monitoring the diversity score enables us to assess whether the model continues to introduce new variability in the generated data, serving as an online signal for generation quality.

If we denote by $S$ the final inference step, then the output of the generation process is the complete embedding table $\tilde{E} \in \mathbb{R}^{\tilde{I} \times K}$, where, for each channel $c$, $\tilde{E}^c = \tilde{E}^{c,\leq S}$.

## B  Evaluation metrics

All the metrics presented below are adopted from Ang et al. (2023), except for *Shapelet-based Reconstruction*, which follows the definition in Zheng et al. (2016).

## B.1 Feature-based evaluation

**Marginal Distribution Difference** MDD computes an empirical histogram for each dimension and time step in the generated series, using the bin centers and widths from the original series. It then calculates the average absolute difference between this histogram and that of the original series across bins, assessing how closely the distributions of the original and generated series align.

**AutoCorrelation Difference** ACD computes the autocorrelation of both the original and generated time series, then determines their difference. By contrasting the autocorrelations, we could evaluate how well dependencies are maintained in the generated time series.

**Skewness Difference** SD is vital for the marginal distribution of a time series, quantifying its distribution asymmetry. Given the mean (standard deviation) of the train time series $T_s^{tr}$ as $\mu_s^{tr}$ ($\sigma_s^{tr}$) and the generated time series $T_s^{gen}$ as $\mu_s^{gen}$ ($\sigma_s^{gen}$), we evaluate the fidelity of $T_s^{gen}$ by computing the skewness difference between them as:

$$SD = \left| \frac{\mathbb{E}[(T_s^{gen} - \mu_s^{gen})^3]}{\sigma_s^{gen3}} - \frac{\mathbb{E}[(T_s^{tr} - \mu_s^{tr})^3]}{\sigma_s^{tr3}} \right|.$$

**Kurtosis Difference** Like skewness, KD assesses the tail behavior of a distribution, revealing extreme deviations from the mean. Using the previous notations, the kurtosis difference between $T_s^{tr}$ and $T_s^{gen}$ is calculated as:

$$KD = \left| \frac{\mathbb{E}[(T_s^{gen} - \mu_s^{gen})^4]}{\sigma_s^{gen4}} - \frac{\mathbb{E}[(T_s^{tr} - \mu_s^{tr})^4]}{\sigma_s^{tr4}} \right|.$$

## B.2 Distance-based evaluation

**Euclidean Distance** For each original series $s^{tr} = (x_1, ..., x_l)$ and its generated $s^{gen} = (y_1, ..., y_l)$, $ED = \sqrt{\sum_{1=1}^{l}(x_i - y_i)^2}$. We take the mean of ED for all series and all samples. Given that the input time series has been preprocessed to fit within the range of $[0, 1]$, ED deterministically assesses the similarity between $s^{gen}$ and $s^{tr}$. It provides a value-wise comparison between the time series.

**Dynamic Time Warping** Given that ED overlooks alignment, we include DTW to capture the optimal alignment between series regardless of their pace or timing. The alignment facilitated by DTW offers insights into the predictive quality of the generated series.

**Shapelet-based Reconstructions** Shapelet based RE is calculated by generated time series using shapelets extracted using shift invariant dictionary learning (SIDL) algorithm (Zheng et al., 2016). Shapelets represent local discriminative patterns present in the time-series data. We learn shift invariant patterns/shapelets on the original time-series dataset and then use the learnt dictionary to reconstruct unseen generated time series. The reconstruction error is calculated between the generated time series and their reconstruction using SIDL.

# C  Datasets & protocol settings

**Table C.1: Overview of the benchmark datasets.** For each dataset, we report its application domain, sampling frequency, number of time series, length statistics, and the type of evaluation (univariate, multivariate, multi-sample) it supports.

| Dataset | Domain | Freq. | Number | Evaluation type | | |
|---|---|---|---|---|---|---|
| | | | | MS | UV | MV |
| Australian Electricity | energy | 30m | 5 | N | Y | Y |
| Appliances | energy | 10m | 1 | N | Y | N |
| Bikesharing | general | 1H | 3 | N | Y | Y |
| Carbon Capture Plant | nature | 2m | 4 | N | Y | N |
| ETTH1 | energy | 1H | 3 | N | Y | Y |
| ECL | energy | 1H | 320 | Y | N | N |
| Exchange Rate | finance | 1D | 8 | N | Y | N |
| NN5 | finance | 1D | 111 | Y | N | N |
| Tourism | general | 1M | 365 | Y | N | N |
| Traffic | transport | 1H | 3 | N | Y | Y |
| Traffic Monash | transport | 1H | 861 | Y | N | N |
| Solar - Weather | nature | 1H | 653 | Y | N | N |
| Rain - Weather | nature | 1H | 386 | Y | N | N |
| Temperature - Weather | nature | 1H | 362 | Y | N | N |

## C.1  Dataset overview

### Energy

- *Australian Electricity* (Godahewa et al., 2021) contains electricity demand data from 5 states in Australia.
- *Appliances*[2] contains house temperature and humidity conditions monitored with a wireless sensor network, and energy data logegd with m-bus energy meters averaged for 10 minutes periods.
- *ETTH1*[3] contains oil temperatures and other covariates of electrical transformers from two stations in China, measured at 15 minutes granularity but hourly aggregated.
- *ECL*[4] contains electricity consumption of 370 points.

### Mobility and Transport

- *Bikesharing*[5] contains the hourly and daily count of rental bikes between the years 2011 and 2012 in the Capital bike share system with the corresponding weather and seasonal information.
- *Traffic*[6] contains observations of the number of vehicles each hour in four different junctions
- *Traffic Monash* (Godahewa et al., 2021) contains hourly road occupancy readings from sensors in the San Francisco Bay area.
- *Tourism* (Godahewa et al., 2021) dataset from, used for the Kaggle Tourism Forecasting competition. This dataset is non-stationary.

### Nature

- *Carbon Capture Plant* (Jablonka et al., 2023) records the emission profiles of "2-amino-2-methyl-1-propanol" (AMP) and "piperazine" (Pz) collected at every 2 minutes interval.
- *Weather* (Godahewa et al., 2021) contains daily time series of four weather variables (rain, mintemp, maxtemp and solar radiation) measured at weather stations in Australia.

### Finance

---

[2] https://www.kaggle.com/datasets/loveall/appliances-energy-prediction
[3] https://github.com/zhouhaoyi/ETDataset
[4] https://www.kaggle.com/datasets/minhnguyendichnhat/ecl-dataset
[5] https://www.kaggle.com/datasets/lakshmi25npathi/bike-sharing-dataset
[6] https://www.kaggle.com/datasets/fedesoriano/traffic-prediction-dataset

- *Exchange Rate* (Godahewa et al., 2021) contains daily exchange rates for currencies of eight countries (Australia, British, Canada, Switzerland, China, Japan, New Zealand and Singapore) between 1990 and 2016. This dataset is non-stationary.
- *NN5 (Daily, Weekly)* (Godahewa et al., 2021) contains cash withdrawal data from ATMs. This dataset combines stationary and non-stationary time series.

## C.2 Protocols

**Multisample setting** For MS data preparation, we sampled $I = 30$ instances, each of length $L = 250$, resulting in a total training sequence of 15,000 timestamps. Standard scaling per timestamp is applied. For evaluation, we generate 100 synthetic instances.

**Univariate setting** For UV data preparation, we used a training sequence of length $L_0 = 2,000$, segmented into $I = 30$ instances of length $L = 250$ using our periodicity-aware segmentation strategy (cf Appendix A.1). Same standard scaling was applied. For evaluation, we generate 100 synthetic instances.

**Multivariate setting** For fine-tuning TTM in the MV setting, we used consecutive sequences of length $L_0 = 5000, 2500$, and $2500$ for training, validation, and testing, respectively. Each set was segmented into $I = 30, 15$, and $15$ instances of length $L = 1120$ using our period-aware segmentation strategy (Appendix A.1). Periodicity was estimated from the training set. Standard scaling per timestamp, computed on the training set, was applied consistently across all splits. For evaluation, we generated 30 instances, matching the size of the training set.

# D  Additional results

**Table D.1: Variance retained across embedding dimensions.** For each dataset, we report the proportion of total variance retained for embedding dimensions $k = 3$, $k = 5$, and $k = 7$ under both FastICA and FPC decompositions.

| Dataset | FICA | | | FPC | | |
|---|---|---|---|---|---|---|
| | k=3 | k=5 | k=7 | k=3 | k=5 | k=7 |
| Appliances | 0.333 | 0.457 | 0.539 | 0.324 | 0.459 | 0.555 |
| Australian Electricity | 0.703 | 0.853 | 0.912 | 0.787 | 0.895 | 0.938 |
| Bikesharing | 0.492 | 0.662 | 0.744 | 0.599 | 0.734 | 0.796 |
| Carbon Capture Plant | 0.773 | 0.905 | 0.951 | 0.843 | 0.933 | 0.965 |
| ETTH1 | 0.536 | 0.686 | 0.775 | 0.635 | 0.754 | 0.823 |
| ECL | 0.809 | 0.941 | 0.971 | 0.991 | 0.997 | 0.999 |
| Exchange Rate | 0.776 | 0.907 | 0.942 | 0.957 | 0.982 | 0.989 |
| NN5 | 0.336 | 0.479 | 0.590 | 0.719 | 0.780 | 0.826 |
| Tourism | 0.869 | 0.954 | 0.977 | 0.986 | 0.995 | 0.998 |
| Traffic | 0.376 | 0.564 | 0.697 | 0.408 | 0.591 | 0.714 |
| Traffic Monash | 0.541 | 0.722 | 0.823 | 0.745 | 0.845 | 0.902 |
| Rain - Weather | 0.449 | 0.605 | 0.725 | 0.558 | 0.684 | 0.781 |
| Solar - Weather | 0.450 | 0.616 | 0.718 | 0.674 | 0.773 | 0.833 |
| Temperature Max - Weather | 0.493 | 0.602 | 0.685 | 0.864 | 0.893 | 0.915 |
| Temperature Min - Weather | 0.390 | 0.517 | 0.611 | 0.777 | 0.824 | 0.858 |

**Table D.2: Filtering statistics for generated embeddings.** Rejection statistics on the *count* variable from the *bikesharing* dataset, averaged across 5 seeds. Each row reports the proportion of generated samples containing missing values, the fraction of samples discarded by the filtering stage, and the average $\ell_2$ norms of the original, accepted, and discarded embedding vectors.

| | NaN% | Discard% | Norms Original (Avg) | Norms Accepted (Avg) | Norms Discarded (Avg) |
|---|---|---|---|---|---|
| SDF-ICA$_3$ | 3.87 | 1.94 | 1.708 | 1.714 | 19.066 |
| SDF-ICA$_5$ | 36.29 | 1.94 | 2.202 | 2.248 | 9.514 |
| SDF-ICA$_7$ | 58.82 | 0.00 | 2.602 | 2.521 | 0.000 |

**Table D.3: Per-dataset similarity results in the multisample setting.** Average normalized similarity scores (feature-based and distance-based) for each dataset and model.

| | Feature-Based | | | | | Distance-Based | | | | |
|---|---|---|---|---|---|---|---|---|---|---|
| | ecl | nn5 | tourism | traffic | weather | ecl | nn5 | tourism | traffic | weather |
| SDF-ICA$_3$ | 0.402 | _0.195_ | 0.268 | 0.330 | _0.204_ | _0.018_ | 0.129 | _0.032_ | 0.086 | 0.137 |
| SDF-FPC$_3$ | 0.576 | 0.323 | 0.594 | _0.293_ | 0.296 | 0.054 | 0.183 | 0.073 | 0.099 | 0.136 |
| TimeVAE | _0.164_ | **0.109** | _0.211_ | **0.143** | **0.174** | 0.063 | 0.139 | 0.074 | 0.143 | 0.125 |
| TimeVQVAE | 0.754 | 0.380 | 0.818 | 0.437 | 0.498 | **0.003** | **0.068** | **0.009** | **0.053** | **0.069** |
| RtsGAN | **0.049** | 0.347 | **0.174** | 0.344 | 0.315 | 0.075 | _0.074_ | 0.110 | _0.059_ | _0.102_ |
| SdeGAN | 0.499 | 0.219 | 0.539 | 0.391 | 0.308 | 0.299 | 0.652 | 0.290 | 0.496 | 0.616 |
| LS4 | 0.860 | 0.319 | 0.849 | 0.470 | 0.400 | 0.720 | 0.681 | 0.639 | 0.926 | 0.707 |

**Table D.4: Per-dataset similarity results in the univariate setting.** Average normalized similarity scores (feature-based and distance-based) for each dataset and model.

| | Feature-Based | | | | | | | Distance-Based | | | | | | |
|---|---|---|---|---|---|---|---|---|---|---|---|---|---|---|
| | appl. | austr. | bike | carbon | etth1 | exch. | traffic | appl. | austr. | bike | carbon | etth1 | exch. | traffic |
| SDF-ICA$_3$ | _0.486_ | **0.122** | **0.162** | _0.197_ | _0.128_ | 0.155 | 0.269 | 0.088 | _0.073_ | _0.077_ | _0.063_ | 0.080 | 0.115 | 0.059 |
| SDF-FPC$_3$ | 0.509 | **0.122** | _0.169_ | 0.345 | **0.107** | 0.287 | **0.213** | 0.085 | 0.101 | _0.077_ | 0.078 | 0.076 | 0.143 | 0.064 |
| TimeVAE | **0.370** | 0.123 | 0.184 | **0.176** | 0.234 | **0.113** | _0.236_ | 0.119 | 0.079 | 0.155 | 0.102 | 0.140 | _0.091_ | 0.178 |
| TimeVQVAE | 0.531 | 0.439 | 0.371 | 0.572 | 0.354 | 0.578 | 0.312 | **0.045** | **0.031** | **0.037** | **0.005** | **0.031** | **0.031** | **0.035** |
| RtsGAN | 0.567 | 0.300 | 0.273 | 0.226 | 0.244 | 0.217 | 0.312 | _0.056_ | 0.098 | 0.084 | 0.085 | _0.072_ | 0.157 | _0.053_ |
| SdeGAN | 0.634 | 0.128 | 0.174 | 0.409 | 0.173 | _0.115_ | 0.249 | 0.456 | 0.623 | 0.831 | 0.703 | 0.833 | 0.469 | 0.765 |
| LS4 | 0.609 | 0.294 | 0.265 | 0.695 | 0.289 | 0.416 | 0.300 | 0.720 | 0.471 | 0.475 | 0.554 | 0.364 | 0.498 | 0.621 |

**Table D.5: Average generation time: baselines** Average time (in seconds) required to generate synthetic univariate time series for the `bikesharing` dataset across three targets: `count`, `temperature`, and `humidity`. We report results for two input sequence lengths: 250 and 500. All models were evaluated under the same computational constraints (`-mem 20G -cores 1+1 -gpu v100`) using a single NVIDIA V100 GPU.

| Length | SDF-ICA$_3$ | SDF-ICA$_5$ | SDF-ICA$_7$ | SDF-FPC$_3$ | SDF-FPC$_5$ | SDF-FPC$_7$ | TimeVAE | TimeVQVAE | RtsGAN | SdeGAN | LS4 |
|---|---|---|---|---|---|---|---|---|---|---|---|
| **250** | 41.9 | 26.8 | 28.8 | 22.0 | 25.4 | 33.1 | 138.1 | 4574.2 | 2055.7 | 3498.9 | 2804.4 |
| **500** | 38.3 | 22.8 | 26.0 | 17.9 | 22.8 | 26.6 | 112.6 | 4401.4 | 3536.3 | 7316.7 | 2378.9 |

**Table D.6: Ablation study: embedding dimension.** Aggregated similarity-based performance across all datasets in the **multisample** and **univariate** setting.

| | | Feature-based | | | | Distance-based | | | Norm. Avg. | |
|---|---|---|---|---|---|---|---|---|---|---|
| | | MDD | ACD | SD | KD | ED | DTW | SHR | Feat. | Dist. |
| **MULTISAMPLE** | SDF-FPC$_3$ | _0.255_ | 2.166 | _1.323_ | 4.299 | 17.749 | 11.921 | 16.537 | 0.616 | 0.609 |
| | SDF-FPC$_5$ | 0.262 | 3.191 | 1.336 | 3.668 | 17.475 | _11.727_ | 22.893 | 0.714 | 0.535 |
| | SDF-FPC$_7$ | 0.264 | 3.534 | 1.500 | 3.560 | 17.710 | **11.652** | 28.068 | 0.787 | 0.655 |
| | SDF-ICA$_3$ | **0.244** | 1.180 | **0.869** | 2.384 | **16.669** | 12.373 | **6.870** | **0.050** | _0.333_ |
| | SDF-ICA$_5$ | 0.261 | _0.782_ | 1.378 | _2.649_ | _16.743_ | 12.238 | _7.731_ | _0.371_ | **0.307** |
| | SDF-ICA$_7$ | 0.265 | **0.589** | 1.964 | 2.963 | 16.900 | 12.031 | 14.195 | 0.576 | 0.362 |
| **UNIVARIATE** | SDF-FPC$_3$ | 0.308 | 1.480 | 0.801 | 1.690 | 19.340 | 12.809 | _5.452_ | 0.736 | 0.469 |
| | SDF-FPC$_5$ | 0.306 | 1.887 | 0.773 | 1.581 | 20.534 | 12.513 | 8.920 | 0.536 | 0.753 |
| | SDF-FPC$_7$ | 0.309 | 2.399 | 0.774 | 1.954 | 20.470 | **12.079** | 10.982 | 0.947 | 0.654 |
| | SDF-ICA$_3$ | _0.306_ | 1.396 | **0.671** | _1.382_ | **18.802** | 12.435 | **4.856** | **0.169** | **0.163** |
| | SDF-ICA$_5$ | 0.306 | _0.867_ | 0.770 | **1.333** | _19.043_ | _12.261_ | 6.555 | 0.279 | _0.222_ |
| | SDF-ICA$_7$ | **0.306** | **0.597** | _0.736_ | 1.458 | 19.989 | 12.381 | 8.102 | _0.175_ | 0.543 |

**Table D.7: Ablation study: embedding dimension.** TTM forecasting performance on downstream tasks using different training sources: generated data, and a combination of original and generated data. Results are reported for 3 multivariate datasets: *bikesharing* (target: `count`, control: `temperature`, `humidity`), *etth1* (target: `HUFL`, control: `MUFL`, `OT`), and *traffic* (target: `junction1`, control: `junction2`, `junction3`). Metrics include RMSE, MASE, WQL, and average rank (lower is better). **Bold** highlights the best result within each row group; **bold+underlined** the overall best.

| | | bikesharing | | | etth1 | | | traffic | | |
|---|---|---|---|---|---|---|---|---|---|---|
| | | RMSE | MASE | WQL | RMSE | MASE | WQL | RMSE | MASE | WQL |
| | 0-shot | 0.728 | 2.150 | 0.287 | 0.678 | 2.132 | 0.255 | 0.708 | **_1.555_** | **_0.255_** |
| | Original Data (OD) | **0.495** | **0.822** | **0.178** | **0.658** | **1.820** | **0.232** | **0.702** | 1.995 | 0.283 |
| GEN | SDF-FPC$_3$ | 0.527 | 0.926 | 0.200 | 0.692 | 1.914 | 0.246 | 0.699 | 2.029 | 0.287 |
| | SDF-FPC$_5$ | 0.530 | 0.918 | 0.198 | 0.693 | 2.003 | 0.252 | 0.662 | **1.837** | **0.262** |
| | SDF-FPC$_7$ | 0.522 | 0.915 | 0.197 | 0.650 | 1.887 | 0.232 | 0.812 | 2.265 | 0.323 |
| | SDF-ICA$_3$ | **0.514** | 0.899 | 0.194 | 0.647 | 1.829 | 0.233 | 0.730 | 2.068 | 0.294 |
| | SDF-ICA$_5$ | 0.537 | 0.909 | 0.194 | 0.637 | 1.934 | 0.233 | **_0.655_** | 1.849 | **0.262** |
| | SDF-ICA$_7$ | 0.517 | **0.898** | **0.193** | **_0.626_** | 1.820 | **_0.224_** | 0.790 | 2.189 | 0.312 |
| OG + GEN | SDF-FPC$_3$ + OD | 0.493 | 0.829 | 0.179 | 0.658 | 1.780 | 0.229 | 0.736 | 2.077 | 0.296 |
| | SDF-FPC$_5$ + OD | 0.487 | 0.807 | 0.174 | 0.659 | 1.757 | 0.230 | 0.743 | 2.087 | 0.297 |
| | SDF-FPC$_7$ + OD | 0.492 | 0.821 | 0.177 | 0.666 | 1.754 | 0.231 | **0.706** | **1.993** | **0.283** |
| | SDF-ICA$_3$ + OD | 0.487 | **_0.801_** | **_0.173_** | **0.640** | 1.790 | 0.228 | 0.734 | 2.074 | 0.295 |
| | SDF-ICA$_5$ + OD | **_0.486_** | 0.804 | 0.174 | 0.649 | 1.780 | 0.230 | 0.750 | 2.110 | 0.301 |
| | SDF-ICA$_7$ + OD | 0.490 | 0.810 | 0.175 | 0.642 | **_1.746_** | **0.226** | 0.718 | 2.025 | 0.288 |

**Table D.8: Average Generation Time Across LLM Backbones.** Average time (in seconds) required to generate synthetic univariate time series for the `bikesharing` dataset across three targets: `count`, `temperature`, and `humidity`. We report results for two input sequence lengths (250 and 500) and compare three LLM backbones: `GPT-2`, `granite-3.0-2b-base`, and `Phi-3.5-mini-instruct`. All models were evaluated under the same computational constraints (`-mem 100G -cores 1+1 -gpu a100`) using a single NVIDIA A100 GPU. For fine-tuning, we use a batch size of 16 for `granite` and 8 for `phi`.

| Length | ICA$_3$ + gpt2 | ICA$_3$ + granite | ICA$_3$ + phi | ICA$_5$ + gpt2 | ICA$_5$ + granite | ICA$_5$ + phi | ICA$_7$ + gpt2 | ICA$_7$ + granite | ICA$_7$ + phi |
|---|---|---|---|---|---|---|---|---|---|
| **250** | 22.7 | 112.8 | 132.6 | 18.3 | 118.5 | 113.9 | 19.0 | 119.9 | 126.1 |
| **500** | 16.2 | 93.6 | 98.5 | 17.3 | 99.9 | 103.7 | 18.9 | 125.0 | 110.7 |

**Table D.9: Ablation study: LLM backbone.** Aggregated similarity-based performance across all datasets for different LLMs used in SDF models. We compare `GPT-2` with two larger and more recent alternatives: `granite-3.0-2b-base`[7] (2B parameters) and `Phi-3.5-mini-instruct`[8] (3.8B parameters). For fine-tuning, we use a batch size of 16 for `granite` and 8 for `phi`.

| | | Feature-based | | | | Distance-based | | | Norm. Avg. | |
|---|---|---|---|---|---|---|---|---|---|---|
| | | MDD | ACD | SD | KD | ED | DTW | SHR | Feat. | Dist. |
| MULTISAMPLE | SDF-FPC$_3$ + GPT-2 | 0.255 | 2.166 | 1.323 | 4.299 | 17.749 | 11.921 | 16.537 | 0.964 | 0.747 |
| | SDF-FPC$_3$ + Granite | 0.251 | 1.817 | 1.227 | 4.132 | _16.429_ | **11.659** | 11.565 | 0.757 | _0.245_ |
| | SDF-FPC$_3$ + Phi-3 | 0.257 | 1.215 | 1.154 | 3.723 | 16.734 | 11.872 | 12.367 | 0.643 | 0.397 |
| | SDF-ICA$_3$ + GPT-2 | _0.244_ | 1.180 | **0.869** | **2.384** | 16.669 | 12.373 | 6.870 | _0.101_ | 0.361 |
| | SDF-ICA$_3$ + Granite | **0.241** | _1.069_ | _0.961_ | _2.524_ | 16.953 | 12.744 | **6.160** | **0.101** | 0.509 |
| | SDF-ICA$_3$ + Phi-3 | 0.247 | **0.907** | 1.102 | 3.570 | **16.069** | _11.847_ | _6.499_ | 0.382 | **0.069** |
| UNIVARIATE | SDF-FPC$_3$ + GPT-2 | 0.308 | 1.480 | 0.801 | 1.690 | 19.340 | 12.809 | 5.452 | 0.947 | 0.804 |
| | SDF-FPC$_3$ + Granite | _0.305_ | **1.268** | 0.673 | _1.185_ | 19.026 | 12.556 | 5.457 | **0.207** | 0.505 |
| | SDF-FPC$_3$ + Phi-3 | 0.310 | _1.368_ | 0.671 | 1.305 | 19.561 | 12.767 | 5.196 | 0.574 | 0.777 |
| | SDF-ICA$_3$ + GPT-2 | 0.306 | 1.396 | _0.671_ | 1.382 | _18.802_ | 12.435 | _4.856_ | 0.496 | _0.161_ |
| | SDF-ICA$_3$ + Granite | **0.304** | 1.370 | 0.679 | **1.123** | 18.616 | 12.365 | 4.712 | _0.253_ | **0.000** |
| | SDF-ICA$_3$ + Phi-3 | 0.307 | 1.398 | **0.541** | 1.199 | 19.081 | 12.471 | 5.856 | 0.337 | 0.577 |

**Table D.10: Multisample evaluation: similarity metrics reported per dataset.**

| | | SDForger Models | | VAE Models | | GAN Models | | Others |
|---|---|---|---|---|---|---|---|---|
| | | ICA$_3$ | FPC$_3$ | TimeVAE | TimeVQVAE | RTSGAN | SDEGAN | LS4 |
| ECL | MDD | 0.154 | 0.219 | 0.156 | 0.292 | **0.145** | 0.193 | 0.296 |
| | ACD | 0.146 | 4.492 | 0.082 | 7.862 | **0.051** | 0.174 | 8.365 |
| | SD | 2.47 | 2.762 | 0.746 | 2.803 | **0.114** | 2.91 | 2.983 |
| | KD | 7.342 | 6.07 | 3.294 | 6.895 | **1.001** | 8.673 | 10.074 |
| | ED | 10.141 | 12.543 | 12.887 | **9.978** | 13.461 | 25.951 | 43.715 |
| | DTW | 9.715 | 10.985 | 12.025 | **7.97** | 12.486 | 25.431 | 40.142 |
| | SHAP-RE | **0.424** | 4.027 | 1.922 | 2.21 | 4.123 | 7.048 | 105.287 |
| NN5 | MDD | 0.248 | 0.248 | **0.243** | 0.371 | 0.383 | 0.246 | 0.262 |
| | ACD | 1.489 | 3.235 | **0.221** | 4.964 | 4.646 | 2.73 | 5.677 |
| | SD | 0.307 | 0.428 | 0.126 | 0.259 | **0.092** | 0.422 | 0.287 |
| | KD | 1.43 | 4.249 | **0.151** | 1.159 | 0.348 | 0.512 | 0.96 |
| | ED | 19.308 | 20.576 | 20.514 | **15.019** | 16.201 | 43.433 | 38.822 |
| | DTW | 12.837 | 12.617 | 11.223 | 10.914 | **9.684** | 36.712 | 24.415 |
| | SHAP-RE | 9.482 | 40.975 | 20.993 | **2.072** | 8.254 | 83.918 | 207.419 |
| Tourism | MDD | 0.189 | 0.24 | 0.172 | 0.339 | **0.121** | 0.208 | 0.282 |
| | ACD | 0.22 | 3.29 | **0.206** | 7.807 | 0.272 | 0.215 | 8.228 |
| | SD | 1.321 | 2.613 | 0.854 | 2.896 | **0.681** | 2.988 | 2.806 |
| | KD | 4.477 | 8.297 | **4.251** | 7.877 | 4.852 | 9.616 | 10.907 |
| | ED | 11.216 | 14.039 | 13.895 | **10.516** | 15.405 | 25.399 | 39.897 |
| | DTW | 10.291 | 11.837 | 12.409 | **8.192** | 14.741 | 25.098 | 35.521 |
| | SHAP-RE | **0.547** | 4.28 | 1.785 | 2.135 | 2.833 | 6.409 | 100.231 |
| Traffic | MDD | 0.251 | 0.234 | 0.234 | 0.359 | 0.314 | **0.222** | 0.242 |
| | ACD | 1.443 | 1.368 | **0.097** | 3.767 | 0.886 | 3.749 | 5.229 |
| | SD | 1.433 | 1.507 | **0.353** | 1.377 | 1.384 | 1.598 | 1.403 |
| | KD | 3.177 | 1.937 | **1.263** | 1.503 | 2.642 | 3.113 | 4.727 |
| | ED | 16.532 | 18.429 | 20.748 | **14.169** | 15.552 | 35.522 | 55.908 |
| | DTW | 10.917 | 10.5 | 12.343 | 10.039 | **9.164** | 31.69 | 37.028 |
| | SHAP-RE | 6.568 | 9.188 | 14.948 | **1.995** | 5.101 | 51.49 | 205.07 |
| Weather (Maxtemp) | MDD | 0.292 | 0.293 | **0.282** | 0.447 | 0.303 | 0.286 | 0.296 |
| | ACD | 1.097 | 2.221 | **0.533** | 6.717 | 1.15 | 0.705 | 7.597 |
| | SD | 0.131 | 0.017 | 0.435 | **0.005** | 0.37 | 0.237 | 0.142 |
| | KD | **0.419** | 2.561 | 0.591 | 2.188 | 0.584 | 0.527 | 0.976 |
| | ED | 25.51 | 21.661 | 19.665 | **15.113** | 19.77 | 43.035 | 46.189 |
| | DTW | 20.532 | 15.354 | 13.238 | **11.219** | 15.884 | 41.331 | 34.55 |
| | SHAP-RE | 6.638 | 24.439 | 9.079 | **2.098** | 3.931 | 34.196 | 152.274 |
| Weather (Mintemp) | MDD | 0.271 | 0.27 | **0.264** | 0.41 | 0.365 | 0.266 | 0.275 |
| | ACD | 1.705 | 1.053 | **0.278** | 5.937 | 1.508 | 1.67 | 6.737 |
| | SD | 0.185 | 0.216 | 0.304 | **0.05** | 0.398 | 0.249 | 0.145 |
| | KD | 1.065 | 1.789 | 0.757 | 1.936 | 0.985 | **0.213** | 1.174 |
| | ED | 17.199 | 18.533 | 19.575 | **15.177** | 15.575 | 44.094 | 46.266 |
| | DTW | 13.148 | 11.255 | **10.48** | 10.675 | 10.837 | 41.651 | 32.415 |
| | SHAP-RE | 6.217 | 14.613 | 15.605 | **1.956** | 2.587 | 55.846 | 208.289 |
| Weather (Rain) | MDD | 0.233 | 0.222 | **0.175** | 0.291 | 0.259 | 0.199 | 0.27 |
| | ACD | 1.642 | 0.321 | **0.309** | 2.491 | 2.735 | 5.549 | 4.696 |
| | SD | **0.599** | 2.474 | 1.065 | 2.683 | 1.641 | 2.729 | 2.135 |
| | KD | **1.146** | 6.636 | 3.238 | 7.94 | 7.447 | 9.639 | 9.724 |
| | ED | 14.461 | 16.009 | 16.979 | **13.631** | 15.033 | 31.412 | 50.966 |
| | DTW | 10.935 | 11.493 | 11.35 | **10.718** | 10.803 | 28.891 | 33.017 |
| | SHAP-RE | 10.955 | 9.555 | 25.464 | **1.883** | 9.595 | 57.118 | 224.024 |
| Weather (Solar) | MDD | 0.314 | 0.31 | 0.29 | 0.459 | 0.342 | 0.297 | **0.284** |
| | ACD | 1.695 | 1.351 | **0.349** | 4.18 | 2.902 | 1.996 | 2.667 |
| | SD | 0.508 | 0.564 | 0.172 | 0.539 | 0.216 | 0.101 | **0.044** |
| | KD | **0.013** | 2.854 | 0.029 | 1.618 | 0.544 | 0.433 | 0.27 |
| | ED | 18.989 | 20.201 | 20.061 | **15.69** | 17.677 | 48.549 | 33.346 |
| | DTW | 10.61 | 11.322 | **9.932** | 11.607 | 11.275 | 36.321 | 17.363 |
| | SHAP-RE | 14.132 | 25.218 | 22.375 | **1.892** | 8.623 | 117.395 | 80.632 |

[7] https://huggingface.co/ibm-granite/granite-3.0-2b-base
[8] https://huggingface.co/microsoft/Phi-3.5-mini-instruct

**Table D.11: Univariate evaluation: similarity metrics reported for Energy datasets.**

| | | SDForger Models | | VAE Models | | GAN Models | | Others |
|---|---|---|---|---|---|---|---|---|
| | | ICA$_3$ | FPC$_3$ | TimeVAE | TimeVQVAE | RTSGAN | SDEGAN | LS4 |
| Appliances | MDD | 0.318 | 0.315 | 0.303 | 0.405 | 0.414 | **0.241** | 0.251 |
| | ACD | 1.825 | **1.714** | 3.137 | **1.714** | 2.404 | 7.208 | 4.301 |
| | SD | 2.068 | 2.093 | **1.008** | 2.07 | 2.152 | 2.269 | 2.028 |
| | KD | 3.249 | 3.816 | 2.891 | 2.586 | **2.559** | 4.196 | 5.874 |
| | ED | 19.493 | 19.13 | 20.459 | **15.916** | 17.162 | 29.562 | 47.81 |
| | DTW | 11.597 | 11.73 | **10.03** | 12.322 | 11.301 | 27.191 | 16.564 |
| | SHAP-RE | 9.588 | 9.072 | 27.262 | **1.952** | 6.367 | 71.477 | 162.749 |
| Australian Elec (T000000) | MDD | 0.315 | 0.312 | 0.292 | 0.457 | 0.376 | **0.281** | 0.323 |
| | ACD | 1.069 | **0.392** | 1.431 | 5.641 | 1.926 | 2.976 | 7.129 |
| | SD | 0.165 | 0.103 | 0.65 | **0.071** | 0.3 | 0.091 | 0.423 |
| | KD | **0.161** | 0.452 | 1.296 | 2.12 | 1.829 | 0.325 | 0.765 |
| | ED | 19.017 | 22.663 | 19.656 | **15.657** | 22.511 | 42.67 | 38.681 |
| | DTW | 10.652 | 13.655 | 10.899 | 10.583 | 14.942 | 40.842 | 26.104 |
| | SHAP-RE | 3.411 | 2.719 | 7.925 | **2.441** | 4.007 | 23.325 | 34.777 |
| Australian Elec (T000001) | MDD | 0.287 | 0.281 | 0.277 | 0.436 | 0.421 | **0.273** | 0.293 |
| | ACD | 1.46 | 0.457 | **0.265** | 5.579 | 4.796 | 2.518 | 6.908 |
| | SD | 0.274 | 0.307 | 0.11 | 0.137 | 0.589 | **0.031** | 0.032 |
| | KD | 1.8 | 1.677 | 0.503 | 1.99 | **0.264** | 0.273 | 0.899 |
| | ED | 18.368 | 19.769 | 19.51 | **15.717** | 20.145 | 44.034 | 37.357 |
| | DTW | 12.248 | 12.842 | **10.312** | 10.648 | 13.322 | 41.201 | 25.029 |
| | SHAP-RE | **2.211** | 2.91 | 6.061 | 2.587 | 3.291 | 26.765 | 30.768 |
| Australian Elec (T000002) | MDD | 0.313 | 0.313 | **0.292** | 0.486 | 0.334 | 0.297 | 0.315 |
| | ACD | 1.196 | 0.569 | **0.237** | 5.675 | 2.541 | 2.559 | 7.209 |
| | SD | 0.42 | 0.718 | 0.322 | 0.598 | 0.605 | 0.387 | **0.229** |
| | KD | **0.179** | 0.69 | 0.539 | 2.111 | 2.386 | 0.389 | 0.804 |
| | ED | 20.624 | 21.174 | 19.537 | **15.889** | 19.896 | 45.934 | 37.731 |
| | DTW | 13.17 | 13.104 | **10.626** | 10.748 | 14.559 | 44.425 | 26.567 |
| | SHAP-RE | 2.683 | 3.04 | 4.984 | 2.38 | **1.572** | 23.29 | 28.057 |
| Australian Elec (T000003) | MDD | 0.283 | 0.289 | 0.279 | 0.444 | 0.41 | **0.261** | 0.286 |
| | ACD | 1.549 | 1.35 | **0.455** | 5.174 | 4.187 | 2.791 | 6.412 |
| | SD | 0.399 | 0.389 | 0.135 | 0.221 | 0.822 | 0.365 | **0.034** |
| | KD | **0.057** | 0.465 | 0.226 | 1.596 | 0.524 | 0.127 | 1.343 |
| | ED | 17.406 | 19.377 | 20.146 | **15.975** | 18.751 | 40.657 | 40.4 |
| | DTW | 11.621 | 14.456 | 11.466 | **11.025** | 11.502 | 37.719 | 28.914 |
| | SHAP-RE | **1.888** | 1.929 | 4.942 | 2.522 | 3.095 | 17.34 | 26.566 |
| Australian Elec (T000004) | MDD | 0.293 | 0.287 | 0.28 | 0.458 | 0.337 | **0.277** | 0.302 |
| | ACD | 1.68 | **0.887** | 1.735 | 4.565 | 2.658 | 3.209 | 6.726 |
| | SD | 0.088 | 0.062 | 0.666 | 0.136 | 0.173 | **0.057** | 0.365 |
| | KD | **0.199** | 0.664 | 1.679 | 2.199 | 0.691 | 0.527 | 0.706 |
| | ED | 19.85 | 19.864 | 20.51 | **15.88** | 19.327 | 44.238 | 41.623 |
| | DTW | 11.8 | 12.582 | **10.921** | 10.933 | 12.972 | 40.349 | 28.13 |
| | SHAP-RE | 4.11 | 4.468 | 11.67 | **2.523** | 4.131 | 39.504 | 54.245 |
| ETTH1 (HUFL) | MDD | 0.326 | 0.32 | **0.29** | 0.476 | 0.381 | 0.291 | 0.326 |
| | ACD | **1.249** | 1.376 | 3.618 | 2.678 | 4.233 | 4.542 | 4.778 |
| | SD | 0.355 | 0.45 | 0.956 | 0.371 | 0.153 | **0.142** | 0.751 |
| | KD | 0.745 | 0.845 | 1.5 | 2.067 | 0.869 | 1.0 | **0.276** |
| | ED | 19.324 | 20.103 | 21.804 | **16.021** | 17.323 | 45.316 | 25.175 |
| | DTW | **9.475** | 9.579 | 10.69 | 11.35 | 10.616 | 38.346 | 9.745 |
| | SHAP-RE | 9.878 | 13.289 | 28.806 | **1.94** | 7.684 | 129.161 | 48.028 |
| ETTH1 (OT) | MDD | 0.235 | 0.24 | **0.225** | 0.354 | 0.268 | 0.254 | 0.272 |
| | ACD | 2.276 | **1.163** | 1.923 | 4.969 | 2.67 | 3.337 | 6.463 |
| | SD | **0.104** | 0.179 | 1.006 | 0.332 | 0.76 | 0.176 | 0.134 |
| | KD | 0.892 | **0.097** | 0.651 | 1.089 | 0.936 | 0.645 | 1.885 |
| | ED | 20.481 | 18.292 | 22.676 | **15.583** | 20.463 | 51.747 | 42.246 |
| | DTW | 12.525 | 12.195 | 12.614 | **10.499** | 11.718 | 49.511 | 25.674 |
| | SHAP-RE | 3.368 | 3.444 | 15.306 | **2.174** | 5.613 | 52.871 | 52.186 |

**Table D.12: Univariate evaluation: similarity metrics reported for Transport datasets.**

| | | SDForger Models | | VAE Models | | GAN Models | | Others |
|---|---|---|---|---|---|---|---|---|
| | | $ICA_3$ | $FPC_3$ | TimeVAE | TimeVQVAE | RTSGAN | SDEGAN | LS4 |
| Bikesharing (Count) | MDD | 0.326 | 0.336 | 0.295 | 0.492 | 0.444 | **0.29** | 0.332 |
| | ACD | 0.856 | **0.691** | 2.493 | 1.712 | 2.451 | 4.306 | 1.972 |
| | SD | 0.101 | 0.12 | 0.202 | 0.042 | 0.251 | 0.61 | **0.035** |
| | KD | 0.571 | 0.258 | 0.125 | 1.447 | **0.091** | 0.218 | 0.387 |
| | ED | 21.389 | 19.406 | 21.314 | **16.172** | 19.806 | 44.217 | 21.848 |
| | DTW | 10.305 | **10.036** | 10.163 | 12.804 | 10.225 | 31.743 | 10.347 |
| | SHAP-RE | 13.665 | 13.77 | 38.651 | **2.012** | 19.352 | 247.88 | 64.166 |
| Bikesharing (Humidity) | MDD | 0.318 | 0.322 | 0.301 | 0.432 | 0.365 | **0.266** | 0.289 |
| | ACD | **1.653** | 1.94 | 4.68 | 2.715 | 3.323 | 4.948 | 3.845 |
| | SD | 0.116 | 0.113 | 0.569 | **0.076** | 0.367 | 0.425 | 0.617 |
| | KD | 0.521 | 0.806 | 0.671 | 1.831 | 0.511 | **0.404** | 1.13 |
| | ED | 19.761 | 18.098 | 25.8 | **16.097** | 18.478 | 44.641 | 51.221 |
| | DTW | 10.699 | 10.847 | 13.416 | 10.907 | **10.25** | 36.872 | 26.846 |
| | SHAP-RE | 4.249 | 4.63 | 43.651 | **2.154** | 7.005 | 88.647 | 100.497 |
| Bikesharing (Temperature) | MDD | 0.37 | 0.372 | 0.363 | 0.451 | 0.386 | **0.291** | 0.339 |
| | ACD | 1.426 | 1.662 | **0.086** | 5.766 | 2.137 | 1.244 | 7.736 |
| | SD | 0.13 | 0.488 | 0.43 | 0.534 | 0.808 | 0.11 | 0.53 |
| | KD | 1.91 | 0.931 | 0.536 | 1.776 | 1.228 | **0.409** | 1.334 |
| | ED | 17.172 | 20.38 | 19.349 | **15.702** | 20.591 | 49.379 | 39.062 |
| | DTW | 12.107 | 12.47 | 11.074 | **10.569** | 15.251 | 46.37 | 26.093 |
| | SHAP-RE | 2.807 | 4.426 | 7.514 | **2.189** | 6.538 | 42.472 | 50.348 |
| Traffic (Junction 1) | MDD | 0.3 | 0.294 | 0.277 | 0.42 | 0.403 | **0.267** | 0.313 |
| | ACD | 0.924 | **0.861** | 4.631 | 2.642 | 2.195 | 6.292 | 3.854 |
| | SD | 0.276 | 0.018 | 0.667 | **0.008** | 0.225 | 0.167 | 0.266 |
| | KD | 3.793 | 2.151 | 0.769 | 1.807 | 0.317 | **0.052** | 1.168 |
| | ED | 18.191 | 18.465 | 25.655 | **15.782** | 16.151 | 45.765 | 36.403 |
| | DTW | 10.776 | 10.597 | 13.931 | 10.893 | **10.065** | 41.767 | 18.485 |
| | SHAP-RE | 5.198 | 5.21 | 31.453 | **2.124** | 4.393 | 82.871 | 58.632 |
| Traffic (Junction 2) | MDD | 0.367 | 0.368 | 0.352 | 0.434 | 0.445 | **0.281** | 0.315 |
| | ACD | 2.893 | **2.265** | 2.889 | 2.878 | 2.516 | 5.164 | 4.492 |
| | SD | 0.272 | 0.237 | 0.526 | **0.086** | 0.992 | 0.328 | 0.295 |
| | KD | 1.212 | 0.645 | 1.135 | 1.921 | 2.667 | **0.273** | 1.155 |
| | ED | 17.937 | 18.765 | 21.622 | **16.007** | 17.114 | 44.67 | 41.478 |
| | DTW | 10.246 | 10.425 | 10.307 | 10.995 | 11.314 | 38.389 | 19.57 |
| | SHAP-RE | 5.546 | 6.777 | 30.016 | **2.157** | 3.426 | 135.497 | 99.919 |
| Traffic (Junction 3) | MDD | 0.326 | 0.326 | 0.326 | 0.41 | 0.361 | **0.25** | 0.276 |
| | ACD | 2.788 | 2.444 | 3.861 | **2.123** | 3.308 | 5.97 | 4.377 |
| | SD | 0.898 | 1.039 | 0.054 | 0.906 | 0.23 | 1.056 | 1.257 |
| | KD | 1.086 | 0.684 | 0.262 | 0.336 | 0.776 | 1.283 | 2.822 |
| | ED | 17.961 | 18.412 | 22.054 | **16.058** | 18.829 | 38.533 | 53.251 |
| | DTW | 11.558 | 11.147 | 10.892 | 11.807 | **10.677** | 32.538 | 25.618 |
| | SHAP-RE | 6.848 | 7.446 | 46.675 | **2.022** | 12.436 | 118.245 | 178.736 |

**Table D.13: Univariate evaluation: similarity metrics reported for Nature datasets.**

| | | SDForger Models | | VAE Models | | GAN Models | | Others |
|---|---|---|---|---|---|---|---|---|
| | | $ICA_3$ | $FPC_3$ | TimeVAE | TimeVQVAE | RTSGAN | SDEGAN | LS4 |
| CCP ($CO_2$) | MDD | 0.173 | 0.186 | **0.166** | 0.3 | 0.179 | 0.232 | 0.236 |
| | ACD | 1.391 | **1.119** | 2.203 | 5.536 | 2.598 | 3.06 | 6.866 |
| | SD | 0.472 | 2.21 | 2.481 | 1.984 | 1.12 | 1.809 | 1.86 |
| | KD | **1.644** | 3.443 | 4.155 | 3.31 | 3.72 | 5.09 | 6.164 |
| | ED | 18.442 | 18.729 | 26.232 | **14.19** | 14.734 | 72.136 | 36.803 |
| | DTW | 15.245 | 14.344 | 18.308 | **10.176** | 10.831 | 71.196 | 28.544 |
| | SHAP-RE | **1.123** | 1.684 | 13.719 | 2.486 | 1.329 | 52.729 | 25.99 |
| CCP ($NH_3$) | MDD | 0.452 | 0.477 | 0.409 | 0.67 | 0.392 | **0.364** | 0.455 |
| | ACD | **0.312** | 2.223 | 0.55 | 6.779 | 0.56 | 0.822 | 9.168 |
| | SD | 0.416 | 0.766 | 0.272 | 0.441 | **0.23** | 0.346 | 0.753 |
| | KD | 1.267 | 0.554 | 0.212 | 0.292 | 0.671 | **0.011** | 2.743 |
| | ED | 16.965 | 19.28 | 17.18 | **15.465** | 22.525 | 31.041 | 35.404 |
| | DTW | 15.495 | 17.18 | 14.483 | 12.381 | 19.973 | 30.589 | 29.251 |
| | SHAP-RE | **0.423** | 1.064 | 1.13 | 2.418 | 2.904 | 2.805 | 9.832 |
| CCP ($C_4H_{11}NO$) | MDD | 0.197 | 0.191 | **0.158** | 0.266 | 0.189 | 0.189 | 0.229 |
| | ACD | 0.662 | 0.822 | **0.152** | 6.672 | 0.371 | 0.231 | 8.524 |
| | SD | 1.747 | 2.232 | **0.007** | 2.294 | 0.626 | 2.729 | 2.511 |
| | KD | 2.054 | 3.638 | 3.105 | 4.998 | **0.251** | 6.454 | 8.179 |
| | ED | 15.92 | 16.128 | 15.448 | **11.295** | 14.54 | 38.679 | 49.348 |
| | DTW | 12.168 | 11.741 | 10.881 | **8.252** | 10.213 | 37.706 | 40.586 |
| | SHAP-RE | **1.255** | 2.426 | 2.801 | 2.363 | 2.042 | 14.476 | 47.176 |
| CCP ($C_4H_{10}N_2$) | MDD | 0.178 | 0.18 | **0.162** | 0.249 | 0.221 | 0.19 | 0.249 |
| | ACD | 1.514 | 1.273 | **0.535** | 4.522 | 2.445 | 3.142 | 5.538 |
| | SD | 1.675 | 1.638 | **0.512** | 1.456 | 1.398 | 1.831 | 1.659 |
| | KD | **0.157** | 2.451 | 0.824 | 3.091 | 4.551 | 4.592 | 6.204 |
| | ED | 15.698 | 17.378 | 19.159 | **13.879** | 21.6 | 46.177 | 45.823 |
| | DTW | 11.646 | 12.004 | 12.098 | **10.002** | 14.647 | 44.366 | 33.614 |
| | SHAP-RE | 2.299 | 4.547 | 6.142 | **2.228** | 5.582 | 31.911 | 59.928 |

**Table D.14: Univariate evaluation: similarity metrics reported for Finance datasets.**

| | | SDForger Models | | VAE Models | | GAN Models | | Others |
|---|---|---|---|---|---|---|---|---|
| | | $ICA_3$ | $FPC_3$ | TimeVAE | TimeVQVAE | RTSGAN | SDEGAN | LS4 |
| Exchange Rate (Currency 1) | MDD | 0.27 | 0.269 | **0.267** | 0.415 | 0.358 | 0.278 | 0.308 |
| | ACD | 0.851 | 1.727 | **0.211** | 7.212 | 1.062 | 0.932 | 8.803 |
| | SD | 0.409 | 0.62 | 0.33 | 0.614 | 1.031 | 0.476 | 0.332 |
| | KD | 0.32 | 1.173 | 0.058 | 1.77 | 0.854 | **0.034** | 1.275 |
| | ED | 18.913 | 19.909 | 20.652 | **15.335** | 26.589 | 47.853 | 35.288 |
| | DTW | 15.672 | 15.91 | 16.319 | **11.256** | 22.394 | 46.856 | 26.695 |
| | SHAP-RE | **1.759** | 2.228 | 2.52 | 2.449 | 5.691 | 17.248 | 22.698 |
| Exchange Rate (Currency 2) | MDD | 0.316 | 0.303 | 0.286 | 0.459 | 0.343 | **0.253** | 0.291 |
| | ACD | 0.426 | 1.424 | **0.305** | 7.14 | 0.983 | 0.976 | 8.597 |
| | SD | 1.285 | 0.914 | 0.289 | 0.863 | 0.424 | 1.034 | 1.083 |
| | KD | 2.003 | 1.249 | 1.507 | 1.576 | 0.443 | **0.208** | 1.636 |
| | ED | 17.727 | 19.579 | 18.253 | **14.43** | 18.272 | 32.215 | 45.617 |
| | DTW | 15.12 | 16.076 | 14.15 | **10.689** | 14.488 | 31.633 | 38.042 |
| | SHAP-RE | **0.967** | 1.186 | 1.744 | 2.442 | 2.217 | 6.681 | 28.819 |
| Exchange Rate (Currency 3) | MDD | 0.369 | 0.428 | 0.36 | 0.614 | 0.47 | **0.314** | 0.348 |
| | ACD | 0.113 | 2.852 | 0.157 | 8.035 | 0.639 | **0.106** | 9.934 |
| | SD | 0.22 | 0.751 | **0.122** | 0.453 | 1.154 | 0.633 | 0.71 |
| | KD | 0.64 | 3.039 | 0.205 | 2.755 | **0.046** | 0.992 | 0.369 |
| | ED | 19.578 | 21.042 | 19.31 | **14.765** | 21.985 | 28.394 | 38.289 |
| | DTW | 19.092 | 19.25 | 18.532 | **12.47** | 20.528 | 28.143 | 34.966 |
| | SHAP-RE | **0.485** | 0.956 | 0.808 | 2.44 | 1.168 | 2.133 | 15.265 |
| Exchange Rate (Currency 4) | MDD | 0.342 | 0.341 | 0.341 | 0.543 | 0.349 | **0.279** | 0.317 |
| | ACD | **0.34** | 1.427 | 0.848 | 7.171 | 1.132 | 1.023 | 8.913 |
| | SD | 0.647 | 0.252 | **0.136** | 0.349 | 0.49 | 0.502 | 0.686 |
| | KD | 0.484 | 1.206 | 0.69 | 2.483 | **0.394** | 0.666 | 0.645 |
| | ED | 20.09 | 20.287 | 19.622 | **15.38** | 20.51 | 33.675 | 42.123 |
| | DTW | 16.021 | 16.187 | 13.507 | **10.985** | 14.745 | 32.558 | 33.075 |
| | SHAP-RE | **1.18** | 1.383 | 1.85 | 2.34 | 1.858 | 7.303 | 23.273 |
| Exchange Rate (Currency 6) | MDD | 0.273 | 0.28 | 0.251 | 0.427 | 0.338 | **0.237** | 0.286 |
| | ACD | 0.204 | 1.893 | **0.179** | 7.773 | 0.578 | 0.414 | 8.341 |
| | SD | 0.285 | 0.755 | 0.245 | 0.342 | **0.049** | 0.481 | 0.464 |
| | KD | 0.534 | 3.95 | 0.88 | 1.701 | 1.127 | **0.152** | 1.401 |
| | ED | 19.809 | 19.763 | 16.956 | **14.579** | 21.034 | 39.025 | 34.643 |
| | DTW | 18.469 | 17.954 | 14.799 | **11.822** | 18.02 | 38.671 | 30.317 |
| | SHAP-RE | **0.569** | 0.82 | 1.036 | 2.386 | 2.427 | 5.158 | 28.882 |
| Exchange Rate (Currency 7) | MDD | 0.318 | 0.353 | 0.322 | 0.493 | 0.374 | **0.292** | 0.334 |
| | ACD | 0.17 | 2.423 | 0.179 | 7.967 | **0.121** | 0.22 | 9.778 |
| | SD | 0.249 | **0.169** | 0.212 | 0.22 | 0.61 | 0.344 | 0.599 |
| | KD | 0.743 | 3.389 | 0.574 | 2.616 | **0.379** | 0.759 | 0.443 |
| | ED | 19.565 | 23.189 | 15.034 | **14.768** | 21.199 | 36.463 | 34.437 |
| | DTW | 18.484 | 20.773 | 13.105 | **12.875** | 19.659 | 36.214 | 30.555 |
| | SHAP-RE | 0.945 | **0.761** | 0.981 | 2.496 | 0.918 | 2.649 | 9.963 |
| Exchange Rate (Currency 8) | MDD | 0.401 | 0.378 | 0.358 | 0.564 | 0.467 | **0.339** | 0.353 |
| | ACD | **0.06** | 2.259 | 0.119 | 8.052 | 0.197 | 0.077 | 10.102 |
| | SD | 0.091 | **0.04** | 0.595 | 0.173 | 0.253 | 0.041 | 0.333 |
| | KD | 1.009 | 2.867 | 0.716 | 2.751 | **0.202** | 0.891 | 0.205 |
| | ED | 16.88 | 21.571 | 16.902 | **14.882** | 21.432 | 37.487 | 42.048 |
| | DTW | 16.254 | 20.245 | 15.651 | **12.884** | 20.061 | 37.232 | 39.402 |
| | SHAP-RE | **0.232** | 1.044 | 0.638 | 2.43 | 0.679 | 3.228 | 15.895 |

# NeurIPS Paper Checklist

1. **Claims**

   Question: Do the main claims made in the abstract and introduction accurately reflect the paper's contributions and scope?

   Answer: [Yes]

   Justification: The abstract and introduction accurately summarize the key contributions and scope of the paper. Specifically, they outline SDForger's novel framework for time series generation using LMs, highlight the use of compact functional embeddings, and emphasize its flexibility, scalability, and ability to work in data-scarce regimes. These claims are substantiated throughout the paper via detailed methodology (Section 3), empirical results (Section 4), and ablation studies (Appendix D), confirming the consistency between the stated objectives and the actual contributions

   Guidelines:

   - The answer NA means that the abstract and introduction do not include the claims made in the paper.
   - The abstract and/or introduction should clearly state the claims made, including the contributions made in the paper and important assumptions and limitations. A No or NA answer to this question will not be perceived well by the reviewers.
   - The claims made should match theoretical and experimental results, and reflect how much the results can be expected to generalize to other settings.
   - It is fine to include aspirational goals as motivation as long as it is clear that these goals are not attained by the paper.

2. **Limitations**

   Question: Does the paper discuss the limitations of the work performed by the authors?

   Answer: [Yes]

   Justification: The paper explicitly discusses its main limitations in the conclusion. In particular, it highlights that the generation quality depends on the embedding dimensionality $k$ and the number of training instances. When $k$ is too large and training data is limited, the fine-tuning of the LM becomes unstable, potentially leading to low-quality or discarded generations. Additionally, the paper acknowledges that model performance may vary depending on dataset complexity and that the optimal $k$ may be task-dependent. The need for a separate filtering step to ensure high-quality output is also discussed. These considerations reflect awareness of both computational and methodological boundaries.

   Guidelines:

   - The answer NA means that the paper has no limitation while the answer No means that the paper has limitations, but those are not discussed in the paper.
   - The authors are encouraged to create a separate "Limitations" section in their paper.
   - The paper should point out any strong assumptions and how robust the results are to violations of these assumptions (e.g., independence assumptions, noiseless settings, model well-specification, asymptotic approximations only holding locally). The authors should reflect on how these assumptions might be violated in practice and what the implications would be.
   - The authors should reflect on the scope of the claims made, e.g., if the approach was only tested on a few datasets or with a few runs. In general, empirical results often depend on implicit assumptions, which should be articulated.
   - The authors should reflect on the factors that influence the performance of the approach. For example, a facial recognition algorithm may perform poorly when image resolution is low or images are taken in low lighting. Or a speech-to-text system might not be used reliably to provide closed captions for online lectures because it fails to handle technical jargon.
   - The authors should discuss the computational efficiency of the proposed algorithms and how they scale with dataset size.

- If applicable, the authors should discuss possible limitations of their approach to address problems of privacy and fairness.
- While the authors might fear that complete honesty about limitations might be used by reviewers as grounds for rejection, a worse outcome might be that reviewers discover limitations that aren't acknowledged in the paper. The authors should use their best judgment and recognize that individual actions in favor of transparency play an important role in developing norms that preserve the integrity of the community. Reviewers will be specifically instructed to not penalize honesty concerning limitations.

3. **Theory assumptions and proofs**

   Question: For each theoretical result, does the paper provide the full set of assumptions and a complete (and correct) proof?

   Answer: [NA]

   Justification: The paper does not include formal theoretical results, theorems, or proofs. The work is empirical and methodological in nature, focusing on the design and evaluation of a generative framework for time-series data using large language models.

   Guidelines:

   - The answer NA means that the paper does not include theoretical results.
   - All the theorems, formulas, and proofs in the paper should be numbered and cross-referenced.
   - All assumptions should be clearly stated or referenced in the statement of any theorems.
   - The proofs can either appear in the main paper or the supplemental material, but if they appear in the supplemental material, the authors are encouraged to provide a short proof sketch to provide intuition.
   - Inversely, any informal proof provided in the core of the paper should be complemented by formal proofs provided in appendix or supplemental material.
   - Theorems and Lemmas that the proof relies upon should be properly referenced.

4. **Experimental result reproducibility**

   Question: Does the paper fully disclose all the information needed to reproduce the main experimental results of the paper to the extent that it affects the main claims and/or conclusions of the paper (regardless of whether the code and data are provided or not)?

   Answer: [Yes]

   Justification: The paper provides detailed information about the datasets, model components (e.g., FPC, ICA, GPT-2), hyperparameters, and evaluation metrics used in both similarity and utility experiments. The experimental setup, including instance length, number of components, fine-tuning strategy, and prompt structure, is clearly described in Section 5 and Appendix D. These details allow for the reproduction of the main results.

   Guidelines:

   - The answer NA means that the paper does not include experiments.
   - If the paper includes experiments, a No answer to this question will not be perceived well by the reviewers: Making the paper reproducible is important, regardless of whether the code and data are provided or not.
   - If the contribution is a dataset and/or model, the authors should describe the steps taken to make their results reproducible or verifiable.
   - Depending on the contribution, reproducibility can be accomplished in various ways. For example, if the contribution is a novel architecture, describing the architecture fully might suffice, or if the contribution is a specific model and empirical evaluation, it may be necessary to either make it possible for others to replicate the model with the same dataset, or provide access to the model. In general. releasing code and data is often one good way to accomplish this, but reproducibility can also be provided via detailed instructions for how to replicate the results, access to a hosted model (e.g., in the case of a large language model), releasing of a model checkpoint, or other means that are appropriate to the research performed.

- While NeurIPS does not require releasing code, the conference does require all submissions to provide some reasonable avenue for reproducibility, which may depend on the nature of the contribution. For example
  (a) If the contribution is primarily a new algorithm, the paper should make it clear how to reproduce that algorithm.
  (b) If the contribution is primarily a new model architecture, the paper should describe the architecture clearly and fully.
  (c) If the contribution is a new model (e.g., a large language model), then there should either be a way to access this model for reproducing the results or a way to reproduce the model (e.g., with an open-source dataset or instructions for how to construct the dataset).
  (d) We recognize that reproducibility may be tricky in some cases, in which case authors are welcome to describe the particular way they provide for reproducibility. In the case of closed-source models, it may be that access to the model is limited in some way (e.g., to registered users), but it should be possible for other researchers to have some path to reproducing or verifying the results.

5. **Open access to data and code**

Question: Does the paper provide open access to the data and code, with sufficient instructions to faithfully reproduce the main experimental results, as described in supplemental material?

Answer: [Yes]

Justification: All datasets used in our experiments are publicly available and listed in the paper. We will include the code as supplemental material at submission time and will release it as open-source upon acceptance.

Guidelines:

- The answer NA means that paper does not include experiments requiring code.
- Please see the NeurIPS code and data submission guidelines (`https://nips.cc/public/guides/CodeSubmissionPolicy`) for more details.
- While we encourage the release of code and data, we understand that this might not be possible, so "No" is an acceptable answer. Papers cannot be rejected simply for not including code, unless this is central to the contribution (e.g., for a new open-source benchmark).
- The instructions should contain the exact command and environment needed to run to reproduce the results. See the NeurIPS code and data submission guidelines (`https://nips.cc/public/guides/CodeSubmissionPolicy`) for more details.
- The authors should provide instructions on data access and preparation, including how to access the raw data, preprocessed data, intermediate data, and generated data, etc.
- The authors should provide scripts to reproduce all experimental results for the new proposed method and baselines. If only a subset of experiments are reproducible, they should state which ones are omitted from the script and why.
- At submission time, to preserve anonymity, the authors should release anonymized versions (if applicable).
- Providing as much information as possible in supplemental material (appended to the paper) is recommended, but including URLs to data and code is permitted.

6. **Experimental setting/details**

Question: Does the paper specify all the training and test details (e.g., data splits, hyperparameters, how they were chosen, type of optimizer, etc.) necessary to understand the results?

Answer: [Yes]

Justification: The paper provides all key experimental details in Section 5 and Appendix C, including data splits, embedding dimensionality, fine-tuning parameters, optimizer, and stopping criteria. Hyperparameter choices are justified through ablation studies, and additional implementation details are included in the supplementary material.

Guidelines:

- The answer NA means that the paper does not include experiments.
- The experimental setting should be presented in the core of the paper to a level of detail that is necessary to appreciate the results and make sense of them.
- The full details can be provided either with the code, in appendix, or as supplemental material.

7. **Experiment statistical significance**

Question: Does the paper report error bars suitably and correctly defined or other appropriate information about the statistical significance of the experiments?

Answer: [No]

Justification: We do not report error bars or use repeated runs with different random seeds. Instead, we demonstrate robustness by evaluating on a wide range of datasets and metrics. We report results for each individual dataset as well as domain-level and overall aggregated scores (e.g., average rank, normalized scores), which helps mitigate dataset-specific variability and highlights consistent trends across diverse temporal settings.

Guidelines:

- The answer NA means that the paper does not include experiments.
- The authors should answer "Yes" if the results are accompanied by error bars, confidence intervals, or statistical significance tests, at least for the experiments that support the main claims of the paper.
- The factors of variability that the error bars are capturing should be clearly stated (for example, train/test split, initialization, random drawing of some parameter, or overall run with given experimental conditions).
- The method for calculating the error bars should be explained (closed form formula, call to a library function, bootstrap, etc.)
- The assumptions made should be given (e.g., Normally distributed errors).
- It should be clear whether the error bar is the standard deviation or the standard error of the mean.
- It is OK to report 1-sigma error bars, but one should state it. The authors should preferably report a 2-sigma error bar than state that they have a 96% CI, if the hypothesis of Normality of errors is not verified.
- For asymmetric distributions, the authors should be careful not to show in tables or figures symmetric error bars that would yield results that are out of range (e.g. negative error rates).
- If error bars are reported in tables or plots, The authors should explain in the text how they were calculated and reference the corresponding figures or tables in the text.

8. **Experiments compute resources**

Question: For each experiment, does the paper provide sufficient information on the computer resources (type of compute workers, memory, time of execution) needed to reproduce the experiments?

Answer: [Yes]

Justification: We describe the compute resources in Appendix D. Experiments were run on NVIDIA V100 GPUs with 20 GB memory and 2 CPU cores. We also include generation time benchmarks (Table D.5) and specify batch sizes, training steps, and stopping criteria. The overall compute budget was moderate, as the models used are lightweight (e.g., GPT-2), and training converges in under 30 minutes for most datasets.

Guidelines:

- The answer NA means that the paper does not include experiments.
- The paper should indicate the type of compute workers CPU or GPU, internal cluster, or cloud provider, including relevant memory and storage.
- The paper should provide the amount of compute required for each of the individual experimental runs as well as estimate the total compute.
- The paper should disclose whether the full research project required more compute than the experiments reported in the paper (e.g., preliminary or failed experiments that didn't make it into the paper).

9. **Code of ethics**

Question: Does the research conducted in the paper conform, in every respect, with the NeurIPS Code of Ethics `https://neurips.cc/public/EthicsGuidelines`?

Answer: [Yes]

Justification: The research fully complies with the NeurIPS Code of Ethics. We use only publicly available datasets that do not contain personal or sensitive information. No human participants or surveillance data were involved. Our method is general-purpose and does not support applications that could lead to discrimination, harm, or misuse. We also discuss potential limitations and plan to release the code under an appropriate license with documentation to support transparency and reproducibility.

Guidelines:

- The answer NA means that the authors have not reviewed the NeurIPS Code of Ethics.
- If the authors answer No, they should explain the special circumstances that require a deviation from the Code of Ethics.
- The authors should make sure to preserve anonymity (e.g., if there is a special consideration due to laws or regulations in their jurisdiction).

10. **Broader impacts**

Question: Does the paper discuss both potential positive societal impacts and negative societal impacts of the work performed?

Answer: [No]

Justification: The paper does not include a dedicated discussion of broader societal impacts, as it focuses on a general-purpose framework for time-series data generation using publicly available datasets. The proposed method does not directly engage with high-risk or sensitive applications, nor does it introduce privacy or fairness concerns in its current scope. We encourage responsible use aligned with the ethical deployment of generative models.

Guidelines:

- The answer NA means that there is no societal impact of the work performed.
- If the authors answer NA or No, they should explain why their work has no societal impact or why the paper does not address societal impact.
- Examples of negative societal impacts include potential malicious or unintended uses (e.g., disinformation, generating fake profiles, surveillance), fairness considerations (e.g., deployment of technologies that could make decisions that unfairly impact specific groups), privacy considerations, and security considerations.
- The conference expects that many papers will be foundational research and not tied to particular applications, let alone deployments. However, if there is a direct path to any negative applications, the authors should point it out. For example, it is legitimate to point out that an improvement in the quality of generative models could be used to generate deepfakes for disinformation. On the other hand, it is not needed to point out that a generic algorithm for optimizing neural networks could enable people to train models that generate Deepfakes faster.
- The authors should consider possible harms that could arise when the technology is being used as intended and functioning correctly, harms that could arise when the technology is being used as intended but gives incorrect results, and harms following from (intentional or unintentional) misuse of the technology.
- If there are negative societal impacts, the authors could also discuss possible mitigation strategies (e.g., gated release of models, providing defenses in addition to attacks, mechanisms for monitoring misuse, mechanisms to monitor how a system learns from feedback over time, improving the efficiency and accessibility of ML).

11. **Safeguards**

Question: Does the paper describe safeguards that have been put in place for responsible release of data or models that have a high risk for misuse (e.g., pretrained language models, image generators, or scraped datasets)?

Answer: [NA]

Justification: The proposed method operates on structured time-series data and leverages publicly available pretrained language models (e.g., GPT-2). It does not release any new pretrained models or scraped datasets, nor does it target high-risk applications. As such, the framework does not pose a significant risk of misuse in its current form.

Guidelines:

- The answer NA means that the paper poses no such risks.
- Released models that have a high risk for misuse or dual-use should be released with necessary safeguards to allow for controlled use of the model, for example by requiring that users adhere to usage guidelines or restrictions to access the model or implementing safety filters.
- Datasets that have been scraped from the Internet could pose safety risks. The authors should describe how they avoided releasing unsafe images.
- We recognize that providing effective safeguards is challenging, and many papers do not require this, but we encourage authors to take this into account and make a best faith effort.

12. **Licenses for existing assets**

Question: Are the creators or original owners of assets (e.g., code, data, models), used in the paper, properly credited and are the license and terms of use explicitly mentioned and properly respected?

Answer: [Yes]

Justification: All datasets and models used in this paper are publicly available and appropriately cited. Where applicable, we have acknowledged the source papers and repositories, Pretrained LMs (such as GPT-2 and Phi) are used under their respective terms of use and clearly referenced in the methodology section.

Guidelines:

- The answer NA means that the paper does not use existing assets.
- The authors should cite the original paper that produced the code package or dataset.
- The authors should state which version of the asset is used and, if possible, include a URL.
- The name of the license (e.g., CC-BY 4.0) should be included for each asset.
- For scraped data from a particular source (e.g., website), the copyright and terms of service of that source should be provided.
- If assets are released, the license, copyright information, and terms of use in the package should be provided. For popular datasets, `paperswithcode.com/datasets` has curated licenses for some datasets. Their licensing guide can help determine the license of a dataset.
- For existing datasets that are re-packaged, both the original license and the license of the derived asset (if it has changed) should be provided.
- If this information is not available online, the authors are encouraged to reach out to the asset's creators.

13. **New assets**

Question: Are new assets introduced in the paper well documented and is the documentation provided alongside the assets?

Answer: [Yes]

Justification: The paper introduces a new framework (SDForger) for time-series generation. The code to reproduce the methodology and results will be submitted as supplemental material and made publicly available under an open-source license upon acceptance. Documentation will be provided to support reproducibility.

Guidelines:

- The answer NA means that the paper does not release new assets.
- Researchers should communicate the details of the dataset/code/model as part of their submissions via structured templates. This includes details about training, license, limitations, etc.

- The paper should discuss whether and how consent was obtained from people whose asset is used.
- At submission time, remember to anonymize your assets (if applicable). You can either create an anonymized URL or include an anonymized zip file.

14. **Crowdsourcing and research with human subjects**

Question: For crowdsourcing experiments and research with human subjects, does the paper include the full text of instructions given to participants and screenshots, if applicable, as well as details about compensation (if any)?

Answer: [NA]

Justification: The paper does not involve crowdsourcing nor research with human subjects.

Guidelines:

- The answer NA means that the paper does not involve crowdsourcing nor research with human subjects.
- Including this information in the supplemental material is fine, but if the main contribution of the paper involves human subjects, then as much detail as possible should be included in the main paper.
- According to the NeurIPS Code of Ethics, workers involved in data collection, curation, or other labor should be paid at least the minimum wage in the country of the data collector.

15. **Institutional review board (IRB) approvals or equivalent for research with human subjects**

Question: Does the paper describe potential risks incurred by study participants, whether such risks were disclosed to the subjects, and whether Institutional Review Board (IRB) approvals (or an equivalent approval/review based on the requirements of your country or institution) were obtained?

Answer: [NA]

Justification: The paper does not involve crowdsourcing nor research with human subjects.

Guidelines:

- The answer NA means that the paper does not involve crowdsourcing nor research with human subjects.
- Depending on the country in which research is conducted, IRB approval (or equivalent) may be required for any human subjects research. If you obtained IRB approval, you should clearly state this in the paper.
- We recognize that the procedures for this may vary significantly between institutions and locations, and we expect authors to adhere to the NeurIPS Code of Ethics and the guidelines for their institution.
- For initial submissions, do not include any information that would break anonymity (if applicable), such as the institution conducting the review.

16. **Declaration of LLM usage**

Question: Does the paper describe the usage of LLMs if it is an important, original, or non-standard component of the core methods in this research? Note that if the LLM is used only for writing, editing, or formatting purposes and does not impact the core methodology, scientific rigorousness, or originality of the research, declaration is not required.

Answer: [Yes]

Justification: LLMs are a core component of the SDForger framework, where they are fine-tuned to generate synthetic time series from structured text prompts. The methodology, implementation details, and the impact of different LLM backbones are thoroughly discussed in Section 3 and Section 5

Guidelines:

- The answer NA means that the core method development in this research does not involve LLMs as any important, original, or non-standard components.
- Please refer to our LLM policy (https://neurips.cc/Conferences/2025/LLM) for what should or should not be described.

