# OpenReview forum: "Forging Time Series with Language: A Large Language Model Approach to Synthetic Data Generation"
_NeurIPS.cc/2025/Conference — NeurIPS 2025 poster_

### Official Review · Reviewer_Exqs · 2025-06-01

**Clarity:** 3
**Significance:** 2
**Originality:** 2
**Rating:** 4
**Confidence:** 3

**Summary:**

The paper proposes SDForger, a synthetic time series generator based on LLMs. The model proceeds as follows: During training, (1) time series are encoded to tabular data, (2) tabular data are encoded to text to tune the LLM; During inference, (1) text is sampled from LLM to decode to tabular data, (2) tabular data is decoded to time series. Extensive experiments have been conducted to validate the method's performance against existing baselines.

**Questions:**

For the first bulletin in the weakness, I hope the author can explain if they believe SDForger can be improved, and if so, what are the possible approaches.

For the questions below, the authors should consider either adding an explanatory paragraph or conducting experiments to address them:
- Can SDForger handle nonstationary time series data? The author should consider adding a corresponding experiment.
- Does embedding time series into tabular data "lose" temporal dependency information? If so, about how much information is "lost"? Would this cause SDForger's performance to degrade when handling strongly correlated sequences?
- The paper highlighted SDForger's "readiness" in the intro, but I did not find an experiment supporting this claim. The author should consider providing some actual prompts generated in the experiment process that can highlight this.
- How do alternative embedding methods, such as AE or TSNE, compare to PCA and ICA? The author should consider adding an explanatory paragraph or conducting an ablation study.
- What is the dimensionality of time series that SDForger can handle?
- How does SDForger handle external covariates associated with the time series data?

**Ethical Concerns:**

["NO or VERY MINOR ethics concerns only"]

**Final Justification:**

In **SDForger readiness**, yes, I was suggesting that the author should include more prompt examples similar to that of Section 6, line 323. Due to the current space limit, I would suggest that the author consider creating a section in the appendix dedicated to illustrating example prompts.

I read through the authors’ rebuttal carefully, and it addresses my concerns satisfactorily. I will revise my evaluation in the author's favor.

**Limitations:**

Negative societal impact: I am concerned about the privacy issue of the model deployment in practice: does SDForger ensure that its generated data does not leak sensitive information from training data?

**Paper Formatting Concerns:**

No.

**Quality:**

2

**Strengths And Weaknesses:**

Strength:
- The paper seems to be clearly written and well structured.
- The technical details seem to be rigorous.

Weakness:
- The empirical performance of the proposed method appears mixed, and the claim that SDForger achieves “consistently strong” and “balanced” performance is less convincing to me.

---

> ### Author Rebuttal · Authors · 2025-07-30
>
> ## Improvement of SDForger
>
> We believe SDForger can be further improved. Its modular design is intentionally built to support **flexible experimentation**, making it easy to explore enhancements or tailor components to specific needs.
>
> We see several promising directions:
> - **Embedding Strategies**: While our current approach relies on linear methods like FastICA and FPC, future work could explore more expressive, nonlinear embeddings (e.g., AE) or multivariate-aware methods like Multivariate FPCA or Multivariate Singular Spectrum Analysis to better capture temporal and inter-channel dependencies.
> - **Parameter-Efficient Fine-Tuning**: We currently use full fine-tuning for the LLM. Incorporating PEFT techniques such as LoRA or adapters could improve scalability, efficiency, and facilitate domain adaptation.
> - **Extended Utility Evaluation**: While we focus on forecasting, SDForger could be evaluated and optimized for broader downstream tasks such as classification or anomaly detection.
> - **Context and Covariate Integration**: By design, SDForger supports integration of external covariates (e.g., categorical or textual data). Expanding this functionality could enable richer conditional generation, and multimodal transfer learning.
>
> In summary, SDForger offers a flexible foundation, and we see meaningful opportunities to improve it both architecturally and in terms of task generalization.
>
> ## Non-stationary datasets
>
> SDForger was evaluated on both stationary and non-stationary datasets. Weather, Bikesharing, and Australian Electricity mostly contain stationary series with clear seasonal patterns.
>
> In contrast, the **currency dataset** is fully non-stationary, reflecting the volatility of financial data. Additionally, **25% of NN5** and **99% of Tourism** series were identified as non-stationary using the **Augmented Dickey-Fuller** (ADF) test. Below are results for three example channels per dataset:
>
> |Dataset|ADF Statistic|p-value|Stationary?|
> |-|-|-|-|
> |currency_1|-1.665|0.4492|No|
> |currency_2|-2.150|0.2250|No|
> |currency_3|-1.353|0.6048|No|
> |NN5-001|-1.767|0.3971|No|
> |NN5-010|-1.958|0.3054|No|
> |NN5-110|-2.387|0.1454|No|
> |Tourism-T000000|-0.934|0.7767|No|
> |Tourism-T000001|-1.531|0.5180|No|
> |Tourism-T000002|-1.219|0.6653|No|
>
> These experiments, coupled with the non-aggregated results in **Table D.7 and D.11**, confirm that SDForger is capable of handling both stationary and non-stationary time series effectively. However, we appreciate the suggestion and will consider adding a dedicated section in the paper to separate results on stationary and non-stationary datasets!
>
> ## Temporal dependency
>
> The embedding methods we use, FastICA and FPC, decompose each fixed-length time series segment into a linear combination of basis components. This strategy retains temporal dependencies, as the components span the entire window and reconstruction preserves their temporal structure.
> The choice of **embedding dimension k** controls the fidelity of this representation and how much information will be “lost”: lower-dimensional embeddings retain less variance, which can lead to some loss of fine-grained temporal or cross-channel details.
>
> Below, we report the **variance retained** by FICA embeddings across datasets:
>
> |data-target|k=3|k=5|k=7|
> |-|-|-|-|
> |appliances - Appliances|0.333|0.457|0.539|
> |bikesharing - cnt|0.633|0.725|0.774|
> |bikesharing - hum|0.424|0.598|0.682|
> |bikesharing - temp|0.418|0.662|0.775|
> |ccp - C4H11NO|0.855|0.946|0.967|
> |ccp - NH3|0.935|0.983|0.997|
> |ccp - CO2|0.697|0.883|0.943|
> |ccp - C4H10N2|0.606|0.808|0.896|
> |ecl|0.809|0.941|0.971|
> |electricity - T000000|0.785|0.922|0.958|
> |electricity - T000001|0.678|0.837|0.894|
> |electricity - T000002|0.779|0.901|0.949|
> |electricity - T000003|0.682|0.818|0.906|
> |electricity - T000004|0.592|0.785|0.855|
> |exchange_rate - currency_1|0.715|0.866|0.916|
> |exchange_rate - currency_2|0.774|0.923|0.952|
> |exchange_rate - currency_3|0.691|0.860|0.912|
> |exchange_rate - currency_4|0.759|0.902|0.943|
> |exchange_rate - currency_6|0.807|0.933|0.962|
> |exchange_rate - currency_7|0.827|0.917|0.946|
> |exchange_rate - currency_8|0.859|0.945|0.965|
> |etth1 - HUFL|0.477|0.636|0.731|
> |etth1 - OT|0.595|0.735|0.819|
> |nn5|0.336|0.479|0.590|
> |monash_traffic|0.541|0.722|0.823|
> |tourism_monthly|0.869|0.954|0.977|
> |weather_maxtemp|0.493|0.602|0.685|
> |weather_mintemp|0.390|0.517|0.611|
> |weather_rain|0.449|0.605|0.725|
> |weather_solar|0.450|0.616|0.718|
> |traffic - Vehicles_Junction_1|0.456|0.657|0.801|
> |traffic - Vehicles_Junction_2|0.379|0.567|0.699|
> |traffic - Vehicles_Junction_3|0.294|0.467|0.590|
>
> As detailed in **Appendix A.2**, we also implement a dynamic strategy to select the number of components based on a target variance threshold, offering practical guidance to adapt \$k\$ to each dataset’s complexity. In practice, when dealing with strongly correlated or complex sequences, using a sufficiently high embedding dimension mitigates information loss and helps maintain SDForger’s performance.
>
> For inter-channel dependencies, we refer to **Q3 of Reviewer V1ut**, where we assess Frobenius distance between real and generated correlation matrices. This ablation confirms that GPT-2 can effectively learn cross-channel structure from compressed multivariate embeddings.
>
> ## SDForger readiness
>
> We kindly ask the Reviewer to clarify whether the question refers to multimodal readiness. If so, this is addressed in **Section 6**, where we demonstrate **text-conditioned generation** by combining structured embeddings with free-form natural language in the prompt. This example shows that SDForger can integrate textual information during both **fine-tuning and inference** without requiring any architectural modifications. The input prompts are presented in **Section 6** (l. 323) and the generated prompts in this experiment are printed in the submitted `conditional_generation.ipynb` notebook (`cell 11`). Due to space constraints, we could not include them in the paper, but we would be happy to share excerpts in a follow-up. This experiment supports our claim of “readiness,” showcasing SDForger’s flexibility for multimodal data. That said, we agree with the Reviewer that **more robust and efficient conditioning strategies** will be important to fully unlock SDForger’s multimodal potential, and we plan to explore this direction in future work.
>
> ## Alternative embedding methods
> In our current study, we focus on classical **linear embedding** method, FastICA and FPC, due to their interpretability, computational efficiency, and well-understood properties in preserving variance, statistical independence, and temporal dependency. Alternative **nonlinear embedding** techniques like Autoencoders (AE) and t-SNE offer powerful representation capabilities and can capture more complex, nonlinear structures in the data. However, they also introduce challenges in terms of:
> - Training **stability** and **complexity**, especially in low-data regimes,
> - **Deterministic reconstruction** quality, which is critical for reliable time series synthesis
> - **Interpretability** and direct control over embedding dimensions.
>
> While t-SNE excels at visualization, it is not designed for reconstructive embeddings, limiting its direct applicability in our generation pipeline. That said, integrating or comparing these alternative embeddings with our LLM-based generation framework represents promising future work. We plan to explore nonlinear embeddings such as AEs in subsequent studies to assess their impact on generation fidelity and downstream task performance.
>
> ## Dimensionality that can be handled
>
> SDForger is not fundamentally limited by the raw length of the time series. Long sequences are split into fixed-length windows using **periodicity-aware segmentation**, and each window is embedded and treated as a separate training instance. Longer sequences thus yield **more examples**, boosting generalization without increasing prompt size.
>
> The main limiting factor is the **embedding dimensionality** $K$, which determines the prompt length. As $K$ increases, prompts grow longer, inference slows down, and generation becomes less stable.
> This holds in **multivariate settings**: more variables imply higher-dimensional embeddings. The embedding size that can be handled depends on the LLM. In our tests, a small model like **GPT-2 handles up to $K = 30$** while producing valid samples.
> Since ICA and FPCA embeddings span entire windows, **window length is not a bottleneck**. The challenge is representing temporal structure with a compact embedding to ensure efficient and stable generation.
>
> ## Handling of external covariates
>
> By design, SDForger supports the integration of **external covariates**, including categorical, numerical, and textual data, into the generation process. These covariates are directly incorporated into the textual prompt, allowing the LLM to condition its generation not only on the time series embeddings but also on rich contextual information.
>
> A concrete example is a weather dataset where each time series channel (e.g., temperature, solar radiation) is associated with metadata about the station, such as geographic coordinates or station ID. These contextual features are encoded as natural language and injected into the prompt, enabling the model to generate sequences that align with both the temporal dynamics and the contextual attributes of the data.
>
> We explore this **conditioning strategy** in **Section 6**, where we demonstrate SDForger’s ability to **generate channel-specific sequences with high fidelity**, identifying the origin of generated curves, indicating that the synthetic sequences effectively preserve contextual distinctions. Integrating richer forms of metadata (e.g. geospatial tags, or event descriptions) will unlock new applications in simulation, personalized forecasting, and beyond.
>
> ## Balanced performances
> We kindly refer to the last paragraph of Reviewer Ck4d rebuttal for a discussion on this.

---

> ### Comment · Reviewer_Exqs · 2025-08-04
>
> In **SDForger readiness**, yes, I was suggesting that the author should include more prompt examples similar to that of Section 6, line 323. Due to the current space limit, I would suggest that the author consider creating a section in the appendix dedicated to illustrating example prompts.
>
> I read through the authors’ rebuttal carefully, and it addresses my concerns satisfactorily. I will revise my evaluation in the author's favor.

---

> > ### Author Response · Authors · 2025-08-05
> >
> > We thank the reviewer for their suggestions and agree that including more prompt examples would improve the clarity of our approach. In the final revision, we will add a dedicated section in the appendix illustrating representative prompt formats used for various datasets and experimental settings.

---

### Official Review · Reviewer_2HPq · 2025-06-22

**Clarity:** 4
**Significance:** 3
**Originality:** 4
**Rating:** 5
**Confidence:** 4

**Summary:**

This paper introduces SDForger, a novel framework for generating synthetic multivariate time series. The core idea is to leverage the generative power of Large Language Models (LLMs) by transforming time series data into a format that an LLM can process. The methodology involves three main stages:
1) An embedding stage where time series instances are projected onto a low-dimensional basis (using FPC or FastICA) to create a tabular representation, decoupling the representation from sequence length.
2) A generation stage where these tabular embeddings are converted into structured text prompts, which are used to fine-tune an autoregressive LLM (e.g., GPT-2). The fine-tuned LLM then generates new textual embeddings.
3) A decoding stage where the generated embeddings are converted back into synthetic time series.
The authors conduct a comprehensive empirical evaluation across 12 datasets, comparing SDForger to several VAE, GAN, and diffusion-based models. SDForger outperforms traditional generative models, such as VAEs, GANs, and diffusion models, in most settings, both in terms of quality and generation speed. The framework is also shown to be multimodal-ready, integrating textual conditioning for greater flexibility.

**Questions:**

**Q1. Clarification on "Data-Scarce" Performance:**
- The paper emphasizes the framework's effectiveness in "data-scarce" settings. However, the experiments use at least I=30 instances for fine-tuning. Could you elaborate on the framework's sensitivity to the number of training instances (I)? How does the generation quality and fine-tuning stability degrade when I is significantly smaller (e.g., I<10)? A clearer understanding of this boundary condition would be valuable for practitioners.

**Q2. Choice of Embedding Method:**
- The results consistently show that the ICA-based variant of SDForger outperforms the FPC-based one, particularly on distance-based metrics. Why might maximizing statistical independence (ICA) be more beneficial for this LLM-based generation pipeline than capturing maximal variance (FPC)? Does this imply that the LLM is better at learning the joint distribution from a set of disentangled factors rather than a hierarchy of variance-ordered components?

**Q3. Generation Rejection Rates:**
- The methodology includes an in-generation filtering step to discard invalid or outlier samples (e.g., those with missing values or diverging norms). Could you report the typical rejection rates observed during your experiments for different datasets or settings? A high rejection rate could signal that the fine-tuned LLM struggles to generate on-manifold embeddings, which would be an important practical limitation to understand.

**Q4. Comparison with Specialized Architectures for Correlation Preservation:**
- The paper compares against general-purpose time series generative models. However, work like COSCI-GAN (NeurIPS 2022) introduces an architecture specifically designed to preserve inter-channel correlations in multivariate time series from a common source, using a central discriminator. How do you position SDForger's implicit method of learning correlations (by encoding a full multivariate instance into a single text prompt) against such an explicit architectural approach? A discussion or comparison in the paper could further highlight the unique strengths of using an LLM for this task.

**Ethical Concerns:**

["NO or VERY MINOR ethics concerns only"]

**Final Justification:**

Based on authors' response to my questions and concerns, I have decided to revise my evaluation in authors' favour.

**Limitations:**

While the authors have done a good job of discussing the technical limitations of their work in the conclusion and the NeurIPS checklist, they have not adequately addressed the potential negative societal impact. In the checklist, the authors state that the work does not engage with high-risk applications. However, synthetic data generation, especially for domains like finance, can have dual-use potential.

**Paper Formatting Concerns:**

None.

**Quality:**

3

**Strengths And Weaknesses:**

- Quality
  - Strengths: The empirical evaluation is extensive and a significant strength of this paper. The authors test their method on a diverse set of 12 public datasets from various domains, which demonstrates broad applicability. The comparison against multiple modern baselines, including VAEs, GANs, and a diffusion model, is thorough. The use of a dual evaluation strategy, comprising both similarity metrics (statistical and distance-based) and a practical utility metric (downstream forecasting), provides a good assessment of the generated data.

  - Weaknesses: The paper claims strong performance in "data-scarce settings," but the experimental setup relies on segmenting a long time series into a minimum number of instances (e.g., I=30). This might not reflect true data scarcity, where only a few short series are available. The statistical robustness of the results could be improved. As acknowledged in the paper's checklist, the experiments do not report error bars or results from multiple runs with different random seeds. While testing on many datasets provides some evidence of robustness, it is not a substitute for assessing variance from stochastic elements like model initialization or data shuffling. The in-generation filtering process, while practical, relies on a heuristic (3 * IQR threshold) to discard outliers, which may not be optimal for all data distributions.

- Clarity
  - Strengths: The paper is well-written and logically structured. Figure 1 provides an excellent, intuitive overview of the entire SDForger pipeline, making the complex methodology much easier to understand. The step-by-step explanation of the embedding, generation, and decoding phases in Section 3 is clear and detailed.

  - Weaknesses: The distinction between the three evaluation settings (multisample, univariate, multivariate) and how data is prepared for each could be introduced more explicitly in the main methodology section for better flow. Also, appendices contain important implementation details that might better serve the reader if summarized in the main text.

- Significance
  - Strengths: The paper addresses a real and growing need for synthetic time series in data-scarce environments. The framework's demonstrated efficiency is a major practical advantage, with generation times being orders of magnitude faster than many competitors (Appendix Table D.3). The most exciting aspect is the framework's extensibility to multimodal generation. The proof-of-concept in Section 6, where generation is conditioned on textual prompts, opens up a wide range of future research and applications in guided and context-aware time series synthesis.

  - Weaknesses: The downstream utility improvements, while present, are sometimes modest. For the traffic dataset, GAN-based methods performed better, suggesting that SDForger may not be universally superior in all scenarios. The long-term impact will depend on how the approach scales to more complex, high-dimensional, and highly non-stationary time series data.

- Originality
  - Strengths: While prior work has used LLMs to generate tabular data, the application to time series through a functional embedding layer is a novel and creative leap. This approach cleverly reframes the sequence generation problem into a structured text generation problem, which is the native domain of LLMs. This is a fundamental departure from the dominant VAE/GAN/diffusion architectures typically used for this task.

  - Weaknesses: The paper builds on established concepts (FPC/ICA, LLM fine-tuning), but their synthesis into the SDForger framework is entirely novel and applied effectively to a new problem domain.

---

> ### Author Rebuttal · Authors · 2025-07-30
>
> ## Q1. Clarification on "data-scarce" performance
>
> We thank the reviewer for raising this important point. In our paper, *data-scarce* refers to both the **limited number of time points** and the **small number of training instances** available for fine-tuning. Unlike most generative approaches requiring thousands of long multivariate sequences, SDForger is designed to operate effectively even when only a **single short time series** is available.
>
> Our **periodicity-aware segmentation** (Section 3.1) enables the extraction of multiple training instances by segmenting the signal into windows aligned with its dominant period. Crucially, the number of training instances $I$ can be adjusted by varying:
>
> * the **window length** $L$: shorter windows yield more segments, and
> * the **window stride**: increasing overlap results in denser sampling.
>
> To assess the **sensitivity to small $I$**, we conducted additional experiments on the **“count” variable from the Bikesharing dataset**, varying both the number of instances and total time points. We fixed the embedding strategy to ICA with $k = 3$, used the same LLM and fine-tuning configuration as in the main paper, and averaged results over **5 seeds**. The table below reports:
>
> * similarity metrics (MDD, ACD, SD, KD, ED, DTW, SHAP-RE),
> * **NaN%**, indicating generations with missing values,
> * discard rate during filtering due to the norm criterion, and
> * average embedding norms (original, accepted, rejected).
>
> | Method    | Instances | Points | MDD   | ACD   | SD    | KD    | ED     | DTW   | SHAP-RE | NaN%  | Discard% | Norms Original (Avg) | Norms Accepted (Avg) | Norms Discarded (Avg) |
> | --------- | --------- | ------ | ----- | ----- | ----- | ----- | ------ | ----- | ------- | ----- | -------- | -------------------- | -------------------- | --------------------- |
> | SDF-ICA-3 | 30        | 1000   | 0.348 | 0.803 | 0.105 | 0.503 | 18.961 | 9.677 | 11.477  | 2.26  | 0.32     | 1.714                | 1.664                | 0.181                 |
> | SDF-ICA-3 | 10        | 1000   | 0.503 | 0.595 | 0.086 | 0.293 | 19.345 | 9.693 | 18.134  | 7.10  | 0.65     | 1.698                | 1.646                | 10.830                |
> | SDF-ICA-3 | 10        | 500    | 0.511 | 0.567 | 0.248 | 0.253 | 19.698 | 9.673 | 16.684  | 12.10 | 2.58     | 1.700                | 1.730                | 0.637                 |
>
> **Key takeaways**:
>
> * SDForger remains **functional even with just 10 instances** and as few as 500 time points.
> * As $I$ decreases, **quality degrades gracefully**, with moderate increases in distance metrics (e.g., MDD, SHAP-RE).
> * **NaN% and discard rates increase**, indicating greater generation instability, but the LLM still produces many valid, high-quality samples.
> * The **norms of accepted embeddings remain close** to those of the real data, confirming that the LLM continues to model the embedding space reliably.
>
> In summary, while extreme scarcity naturally impacts stability, SDForger remains robust and practical even in low-resource settings, with graceful degradation and built-in filtering mechanisms to preserve sample quality.
>
>
> ## Q2. Choice of embedding method
>
> The superior performance of the ICA-based variant likely comes from the nature of the components it produces. Unlike FPC, which orders components by explained variance and often concentrates most information in the first few components, ICA explicitly seeks statistically independent components. This tends to produce a more balanced and disentangled basis decomposition, where each component carries distinct information that have similar importance for data reconstruction. For our LLM-based generation pipeline, this disentanglement appears advantageous because the model can learn a joint distribution over a set of factors that all have the same “power” and are more interpretable. Thus, the results suggest that the LLM is indeed better equipped to model the joint distribution when presented with independent factors rather than a hierarchy of variance-ordered components. This aligns with the broader literature in representation learning, where disentangled representations often lead to improved generative performance and interpretability.
>
>
> ## Q3. Generation rejection rates
>
> #### **Observed Rejection Rates Across Embedding Dimensions**
>
> The table below reports rejection statistics on the **"count" variable from the Bikesharing dataset**, averaged across 5 seeds. We vary the embedding dimension $K$ while keeping all other settings fixed:
>
> | Method    | NaN%  | Discard% | Norms Original (Avg) | Norms Accepted (Avg) | Norms Discarded (Avg) |
> | --------- | ----- | -------- | -------------------- | -------------------- | --------------------- |
> | SDF-ICA-3 | 3.87  | 1.94     | 1.708                | 1.714                | 19.066                |
> | SDF-ICA-5 | 36.29 | 1.94     | 2.202                | 2.248                | 9.514                 |
> | SDF-ICA-7 | 58.82 | 0.00     | 2.602                | 2.521                | 0.000                 |
>
> We observe that:
>
> * **Discard% remains consistently low** (<2%) across settings, indicating that most generations fall within a plausible norm range.
> * **NaN% increases with larger embedding dimensions**, reflecting a higher likelihood of incomplete generations in longer textual outputs. However, these are detected and filtered early, preserving only valid samples.
> * Importantly, **norms of accepted samples closely match the originals**, while rejected samples exhibit extreme values, justifying their removal.
>
> #### **Impact of Filtering on Generation Quality**
>
> We further evaluated the effect of filtering on similarity metrics for $K=3$, again using the "count" variable from the Bikesharing dataset and reporting average results over 5 random seeds.
>
> |Method|Filter|MDD|ACD|SD|KD|ED|DTW|SHAP-RE|NaN%|Discard%|NormsOriginal(Avg)|NormsAccepted(Avg)|NormsDiscarded(Avg)|
> | -| -| -| -| -| -| -| -| -|-| -| -| -| -|
> |SDF-ICA-3|Yes|0.356|0.826|0.100|0.616|19.449|9.907|13.393|3.87|1.94|1.708|1.714|19.066|
> |SDF-ICA-3|No|0.355|0.842|0.209|4.791|20.071|10.932|18.833|-|-|-|-|-|
>
> Filtering leads to a notable improvement in **higher-order statistics**, especially **Skewness (SD)** and **Kurtosis (KD)**, by removing rare but impactful outliers. These outliers, while small in number, disproportionately affect the quality and stability of the generated dataset. Overall, the observed **rejection rates are low**, and the **LLM generates on-manifold embeddings in the majority of cases**, even under tight data constraints. The filtering step acts as a safeguard, eliminating extreme outliers without undermining generation diversity or coverage.
>
> More on the application of similar filtering procedures to alternative generative models (e.g., TimeVAE) is discussed in our response to **Q4 from Reviewer Ck4d.**
>
> ## Q4. Comparison with specialized architectures for correlation preservation
>
> COSCI-GAN and similar specialized architectures explicitly incorporate mechanisms, such as a central discriminator, to explicitly preserve inter-channel correlations in multivariate time series. These designs reflect a targeted approach to model dependencies within the data and have demonstrated strong performance on correlation preservation.
>
> In contrast, SDForger implicitly learns the joint distribution of multivariate time series by encoding an entire instance, including all channels and temporal dynamics, into a single, unified textual prompt.
> This approach offers several unique advantages:
>
> * **Flexibility and Generality**: Rather than engineering architectural components explicitly for correlation modeling, SDForger exploits the powerful contextual modeling capabilities of LLMs. LLM naturally captures complex, high-dimensional dependencies through its attention mechanisms and deep representations, without requiring hand-crafted modules.
> * **Multimodality**: By representing multivariate embeddings as text, SDForger enables the use of off-the-shelf LLMs pre-trained on large, diverse corpora, unlocking transfer learning benefits and opening up opportunities for multimodal extensions, like text-guided generation.
> * **Modularity**: SDForger’s design is inherently modular. While this work uses basis decomposition methods (e.g., FastICA) for general-purpose embedding, the framework allows for the integration of domain-specific or correlation-preserving embeddings. By coupling such embeddings with the LLM’s ability to model joint patterns in long textual sequences, future versions of SDForger could explicitly enhance inter-channel correlation preservation, combining explicit inductive biases with implicit sequence modeling.
>
> While explicit architectures like COSCI-GAN excel in focused correlation preservation, SDForger’s implicit approach via LLMs provides a complementary and versatile framework that can generalize across tasks and modalities. We agree that a clearer discussion of these trade-offs would strengthen the paper.
>
> Additionally, in our response to **Q3 from Reviewer V1ut**, we empirically assess SDForger’s ability to generate consistent multivariate sequences. Results show that our method can effectively preserve channel-wise structure even in low-data regimes.
>
> ## Limitations on potential societal impact and privacy considerations
> We appreciate the reviewer’s important observation regarding the dual-use potential of synthetic data generation in **high-risk applications**. By relying on basis decomposition and text-conditioned generative modeling, SDForger does not memorize or directly reproduce individual time series samples. We acknowledge that synthetic data models could potentially be misused in sensitive domains such as healthcare or finance. We will update the manuscript to reflect these societal and ethical considerations.

---

> > ### Comment · Reviewer_2HPq · 2025-08-04
> >
> > First of all, I want to thank the authors for their prompt response to my questions. They have answered my questions properly.
> >
> > I strongly believe they should clarify the discussed point in the final revision of their paper, especially, a clearer discussion of the mentioned trade-offs about explicit architectures like COSCI-GAN that would definitely strengthen the paper.
> >
> > Conditional on applying these clarifications, I will revise my evaluation in the authors' favour.

---

> > > ### Author Response · Authors · 2025-08-05
> > >
> > > We thank the Reviewer for their comments. We agree that a discussion on the trade-offs between implicit correlation modeling in SDForger and explicit architectures such as COSCI-GAN would enhance the clarity and positioning of our method.
> > >
> > > In the final revision, we will add a dedicated paragraph contrasting SDForger’s modular, text-based generation pipeline with architectures explicitly designed to preserve inter-channel dependencies, highlighting both the strengths and limitations of each approach.

---

> > > > ### Comment · Reviewer_2HPq · 2025-08-05
> > > >
> > > > Thanks for your confirmation. I have revised my evaluation and updaed my rating to `5:Accept` in favour of authors as my concerns have been addressed.

---

### Official Review · Reviewer_Ck4d · 2025-07-01

**Clarity:** 3
**Significance:** 3
**Originality:** 3
**Rating:** 4
**Confidence:** 4

**Summary:**

This paper introduces a method for generating synthetic time series data leveraging LLMs called SDForger. SDForger converts time series data into a tabular representation through basis decomposition techniques (FastICA/FPC). These embeddings are then encoded in a text format for fine tuning of the LLM. Inference follows the inverse path and samples embeddings through text prompts from the LLM. These are then converted to the tabular format and finally back into time series by using the reverse process of the embedding technique. Additionally, the authors use a filtering heuristic on the tabular embeddings to improve quality of the synthetic time series.

The authors evaluate the synthetic time series generated by SDForger by employing similarly based metrics and compare with different generative baselines. In addition, the authors evaluate whether the synthetic data can be used to fine-tune downstream forecasting models. In their ablation study, the authors evaluate different choices of embedding dimensions, inference time, and different choices of the underlying LLM. Finally, the authors conduct a case study on one dataset to illustrate that the method could be further developed into a multi-modal synthetic data generation model.

**Questions:**

**Major**:

How exactly are the embedding dimensions chosen in the utility-based experiments? Do precisely does “dataset complexity” map to these specific values? How would the results change if the embedding dimension is changed for each dataset? This could be an interesting ablation.

What part of the datasets in the utility-based evaluation are used for fine-tuning the LLM and for selecting the embedding dimension? Has the test part been excluded from training/tuning steps? If not, this could lead to data leakage and skew the results.

How do SDFormer perform without the heuristic filtering procedure? I think this could be another interesting dimension and it would be interesting to see how much the performance changes when this would be removed. Additionally, I would be curious to see how much such a filtering step changes the results for TimeVAE. I understand that the filtering is done on the tabular and might not be applicable to TimeVAE, but I think it should be possible to apply diversity/divergent filters here as well.


**Minor**:
What does “balanced performance across all metrics and domains” in the similarity-based evaluation mean specifically? Can this be quantified?

**Ethical Concerns:**

["NO or VERY MINOR ethics concerns only"]

**Final Justification:**

I updated my rating after author discussion see comments.

**Limitations:**

The main limitation of this work lies in the empirical design in the utility-bases section that only evaluates three datasets. This should be noted in the limitation section because it is hard to judge whether the results presented here would generalize because of the small number of evaluation datasets used.

**Quality:**

3

**Strengths And Weaknesses:**

The paper presents a sound and creative framework for generating synthetic time series data. This is of high significance because large scale pre-trained time series models might be bottlenecked through the availability of large datasets which stifles progress. The paper introduces an interesting method for synthetic time series dataset generation by leveraging LLMs through embedding techniques from time series to tabular to text and back. This methodology enables further research on embedding techniques and LLM fine tuning. The paper is written clearly and is easy to follow.

The weaknesses of the paper lie in the empirical evaluation of their method. I understand that synthetic data can be challenging to evaluate. The similarity based results use 12 datasets and several baseline methods. Here, it seems that TimeVAE is still a strong baseline method and that SDForger does not offer significant improvements. On the univariate setting it seems to perform well, but on the multisample setting it does not seem to improver meaningfully over baselines. Given the potential of the method for further development, I do not see this as an issue that would prevent publication, but I would kindly ask the authors to reflect that finding in their writing. Right now, the writing appears justifies SDFormers performance with “balanced performance across all metrics and domains” which I find too vague.

The utility based evaluation has the limitation that only three datasets have been considered and therefore the empirical results have low empirical power. My main concern with the utility based evaluation is that the setup is not quite clear which might impact the validity of the result. Another weakness is that the effect of the filtering heuristic is not shown.

I will detail my specific questions in the Questions section below. I will consider increasing my score if my major questions are addressed.

---

> ### Author Rebuttal · Authors · 2025-07-30
>
> ## Q1: Choice of embedding dimensions
> In our utility-based experiments, we evaluated multiple values of the embedding dimension k across datasets. The values presented in the main paper reflect the best-performing configuration for each dataset in terms of downstream forecasting performance. Indeed, we observe a clear relationship between **dataset complexity** and the **optimal embedding dimension**. As discussed in our response to reviewer V1ut, more complex datasets (traffic and etth1), benefit from higher-dimensional embeddings, that allow **both better reconstruction** and a better learning of **inter-variables correlations**.
>
> We present the utility evaluation results for embedding 3, 5, 7 in the ablation below:
>
> |Dataset|Model|RMSE|MASE|WQL|
> |-|-|-|-|-|
> |bikesharing|0-shot|0.728|2.150|0.287|
> |bikesharing|Original Data (OD)|0.495|0.822|0.178|
> |bikesharing|SDF-ICA-3|0.514|0.899|0.194|
> |bikesharing|SDF-ICA-5|0.537|0.909|0.194|
> |bikesharing|SDF-ICA-7|0.517|0.898|0.193|
> |bikesharing|SDF-ICA-3 + OD|0.487|**0.801**|**0.173**|
> |bikesharing|SDF-ICA-5 + OD|**0.486**|0.804|0.174|
> |bikesharing|SDF-ICA-7 + OD|0.490|0.810|0.175|
> ||||||
> |etth1|0-shot|0.678|2.132|0.255|
> |etth1|Original Data (OD)|0.658|1.820|0.232|
> |etth1|SDF-ICA-3|0.647|1.829|0.233|
> |etth1|SDF-ICA-5|0.637|1.934|0.233|
> |etth1|SDF-ICA-7|**0.626**|1.820|**0.224**|
> |etth1|SDF-ICA-3 + OD|0.640|1.790|0.228|
> |etth1|SDF-ICA-5 + OD|0.649|1.780|0.230|
> |etth1|SDF-ICA-7 + OD|0.642|**1.746**|0.226|
> ||||||
> |traffic|0-shot|0.708|**1.555**|**0.255**|
> |traffic|Original Data (OD)|0.702|1.995|0.283|
> |traffic|SDF-ICA-3|0.730|2.068|0.294|
> |traffic|SDF-ICA-5|**0.655**|1.849|0.262|
> |traffic|SDF-ICA-7|0.790|2.189|0.312|
> |traffic|SDF-ICA-3 + OD|0.734|2.074|0.295|
> |traffic|SDF-ICA-5 + OD|0.750|2.110|0.301|
> |traffic|SDF-ICA-7 + OD|0.718|2.025|0.288|
>
> These results reinforce the embedding selection (realised on validation data) presented in our paper.
> Furthermore, we discuss in **Appendix A.2** a **variance-explained heuristic** that could be used to dynamically select k for a given dataset based on the percentage of variance to retain.
>
> ## Q2: Data leakage?
>
> We confirm that only the training portion of each dataset is used for both fine-tuning the LLM and selecting the embedding dimension. The test set is **strictly held out** and is never used during LLM training, embedding selection, or any intermediate tuning step. As explained, this ensures that there is no data leakage nor biased evaluation between the synthetic data generation process and the evaluation phase. Embedding dimensions  k=3,5,7 are evaluated using validation data (i.e., the 20% subset of the training set as explained in **Section 5**), and the best-performing configuration was selected.
>
> ## Q3: Heuristic filtering procedure
>
> As discussed in our responses to **Q1 and Q3 from Reviewer 2HPq**, SDForger includes an **in-generation filtering step** to discard invalid or outlier samples, specifically those with missing values (**NaN%**) or abnormally large total norms (**Discard%**), computed in the embedding space. This lightweight mechanism improves sample quality and stabilizes distributional properties with minimal overhead.
>
> In the results reported in **Q3**, we show that:
>
> * Across different embedding dimensions, **Discard% remains consistently low (<2%)**, indicating that the LLM generates mostly on-manifold samples.
> * Rejected samples have norms several orders of magnitude higher than accepted ones (e.g., 19.1 vs. 1.7), which strongly impact higher-order statistics such as **skewness** and **kurtosis**.
> * The filtering step notably improves metrics like SD and KD without significantly affecting lower-order distances, showing its value in controlling rare but impactful outliers.
>
> To extend this analysis, we implemented a **post-generation filtering step for TimeVAE**, where we discard generated samples **based on the norm of the reconstructed time series** (in the time domain). Specifically, we constructed SDForger-like intervals and removed curves whose norms fall outside the selected interquartile range $[Q1 - 3 \cdot \text{IQR}, Q3 + 3 \cdot \text{IQR}]$, with bounds computed from the original (real) samples.
>
> The results below (Bikesharing, “count”, 5 seeds) show the effect of this post-hoc filtering:
>
> | Method                | MDD   | ACD   | SD    | KD    | ED     | DTW   | SHAP-RE | Discard% | Norms Original (Avg) | Norms Accepted (Avg) | Norms Discarded (Avg) |
> | --------------------- | ----- | ----- | ----- | ----- | ------ | ----- | ------- | -------- | -------------------- | -------------------- | --------------------- |
> | TimeVAE               | 0.313 | 0.209 | 1.117 | 0.279 | 20.512 | 9.383 | 20.646  | -        | -                    | -                    | -                     |
> | TimeVAE + post filter | 0.316 | 0.220 | 0.131 | 0.064 | 19.601 | 9.182 | 20.689  | 1.211    | 15.752               | 13.473               | 8.916                 |
>
> We observe that:
>
> * **Only a small fraction of samples (\~1.2%) are discarded**, yet this leads to **improvements in SD and KD**, indicating that **a few outliers can disproportionately affect distributional metrics**.
> * This supports the value of post-hoc validation, especially in models like TimeVAE that do not natively include such constraints.
>
> However, we would like to highlight that while post-hoc filtering is a useful safeguard, **embedding lightweight validation directly into the generation process, as done in SDForger, offers both practical and performance advantages**. It enables real-time quality control, avoids unnecessary decoding, and ensures robustness without additional overhead.
>
>
> ## Q4: Clarification on "balanced performances"
>
> In our similarity-based evaluation (Table 1), different generative models exhibit strengths in distinct types of metrics: For example, TimeVAE performs very well on **distribution**-based metrics. Conversely, TimeVQVAE excels on **distance**-based metrics (euclidian distance, dtw ...). In contrast, SDForger achieves a more balanced performance across both metric categories, meaning it consistently **performs well on both feature-based and distance-based metrics** **without heavily favouring one domain**. To quantify this, we computed the normalised average score per metric category per dataset in Table D.2.
>
> This balanced performance indicates that SDForger not only preserves statistical features but also captures distributional similarities effectively, contributing to more robust and versatile synthetic time series generation. We will emphasize this in the updated version of the paper.
>
> We hypothesize that for downstream tasks such as forecasting, it is crucial to use synthetic data generated by models that do not overly prioritize a single metric type but instead achieve consistently **strong performance across multiple metric categories and domains**.

---

> > ### Comment · Reviewer_Ck4d · 2025-08-05
> >
> > Thank you for your rebuttal and the interesting additional experiments on the filtering heuristic and embedding dimension. I still find the distribution/distance-based metrics difficult to judge, even with the updated clarification.
> >
> > As mentioned in my review, one limitation of this study is that the downstream forecasting evaluation only uses three datasets and hence has low empirical power. This has not been addressed and I find this point critical. Thus, I will maintain my score.
> >
> > Overall,  I maintain my positive evaluation of the paper and recommend acceptance

---

> > > ### Author Response · Authors · 2025-08-05
> > >
> > > We thank the reviewer for their feedback and their recommendation for acceptance.
> > > We prioritized a focused evaluation on three datasets coming from diverse domains (industry, transport, nature) in this study.
> > > We agree that expanding the utility evaluation to a broader set of datasets and downstream tasks is a key direction for future work and will strengthen the empirical power of our claims. We will add a note in the revised version highlighting this point explicitly.
> > > Moreover, additional experiments on the filtering heuristic and embedding dimension will be added to the appendix to provide further clarification.

---

### Official Review · Reviewer_V1ut · 2025-07-03

**Clarity:** 2
**Significance:** 2
**Originality:** 3
**Rating:** 4
**Confidence:** 3

**Summary:**

The paper introduces a novel framework for generating synthetic multivariate time series using large language models. The proposed pipeline includes transforming time series into a tabular-structured prompt, where each column contains time series components from FastICA or PCA. These prompts are then used to fine-tune an LLM, which generates new synthetic embeddings that retain the statistical and temporal properties of the original data. This work validates the proposed method through extensive experiments, demonstrating its robustness and applicability in data-scarce settings.

**Questions:**

1. Line 106: How is "natural cycles" determined? Are there some dataset-specific configurations?

2. How about the method's efficiency? Especially, the cost of converting tabular data into text representations. It seems to lead to a very long context.

3. Can the author prove experiments that a structured transformation (using tabular format) effectively generates multivariate time series data?

**Ethical Concerns:**

["NO or VERY MINOR ethics concerns only"]

**Final Justification:**

I acknowledge the efficiency of the proposed method and its performance in multivariate generation. While I no longer have any significant doubts, I believe that the true value of this work may be more apparent when applied to larger LLMs. Overall, I will maintain my positive evaluation.

**Limitations:**

Yes

**Quality:**

2

**Strengths And Weaknesses:**

**Quality**: The submission is technically sound, with claims supported by experimental results across multiple domains (e.g., energy, finance, transport). The authors compare SDForger against multiple baselines (VAEs, GANs, diffusion models) using both similarity-based metrics and utility-based evaluations (forecasting). However, the results could gain more robustness by reporting the error bars and results on more datasets (e.g., long-term forecasting datasets from TSLib and probabilistic forecasting datasets from GIFT-Eval).

**Clarity**: The paper is generally well-structured, but there are areas where clarity could be improved. (1) Abstract: The motivation of the work and the background it aims to solve are not clearly stated here (time series generation, forecasting, or multimodal modeling?). (2) Periodicity-aware segmentation: How is "periodicity-aware" reflected？ (3) LLM fine-tuning: How is the LLM  fine-tuned? Do the authors adopt some PEFT methods?

**Significance**: This paper presents contributions that are likely to influence researchers working with time-series data synthetics. SDForger often can match or exceed GANs/VAEs in similarity and utility metrics. One of the biggest concerns in this work is that the improvement achieved by using this method is not obvious (Tables 1 and 2). I think the proposed method could have a large impact by extending it to other LLM architectures and demonstrating its effectiveness.


**Originality**: I think the work is interesting because the paper provides new insights that LLMs can better understand time series via structured embeddings. While prior work (e.g., TimeGAN, TimeVAE) focused on specialized architectures (GANs, VAEs). This work bridges time-series generation and multimodal models. Technically, I think the author could conduct a comparison using non-structured transformation (e.g., w/o tabular structure or w/o PCA decomposition). This will enhance the persuasiveness of this work in terms of methodology.

---

> ### Author Rebuttal · Authors · 2025-07-30
>
> ## 1. Question on natural cycles
>
> The term **"periodicity-aware"** refers to our method of aligning segmentation windows with the **dominant periodic patterns** of each time series channel. This is done in a fully data-driven way, without requiring dataset-specific configurations.
>
> Specifically, for each channel, we compute the **Autocorrelation Function (ACF)** and identify statistically significant peaks. We select the most prominent peak corresponding to a period $P < L/2$, where $L$ is the target window length. This peak defines the **natural cycle** of the signal.
>
> We then segment the time series into windows of length $L$, using a **step size that is a multiple of $P$**. This ensures that segment boundaries align with intrinsic periodic patterns, improving both **representativeness** and **diversity** of the resulting training instances.
>
> This segmentation strategy enables meaningful augmentation from a single sequence, adapts dynamically to each channel's structure, and **avoids any manual tuning** or domain-specific heuristics.
>
> A detailed and mathematical description of this procedure is provided in **Appendix A.1**.
>
>
> ## 2. Method's efficiency
>
> We address efficiency in **Section 5.3**, with detailed benchmarks in **Appendix Table D.3**, where we report generation times on the **Bikesharing dataset** using different window lengths and embedding sizes.
>
> The efficiency of SDForger is primarily influenced by two factors: the **embedding dimension $K$** and the **underlying LLM architecture**. As $K$ increases, the **length of the generated output** also increases. This affects inference time, as longer sequences take more time to produce, and it can **impact generation quality**, leading to a higher number of discarded samples and requiring more iterations to reach the desired number of valid outputs.  However, even at higher $K$, SDForger remains **highly efficient**, consistently outperforming all the baselines by a wide margin (from 4 to 100 times faster).
>
> The choice of LLM also plays a role. Models like **Granite** require more memory and computation time, typically a **GPU** for inference, but **still remain significantly faster and more efficient** than generative baselines such as diffusion models, especially in low-data regimes.
>
> The **data-to-text conversion step itself is negligible** in terms of overhead. For example, converting an embedding table of size **10 × 30 instances** takes **less than 0.002 seconds** on a **Mac M1 Max**. This step is efficiently handled by the `data_to_text` function in the **`SDForgorger.py`** file.
>
> Overall, SDForger offers a **favorable trade-off between expressiveness and efficiency**, benefiting from compact embeddings and fast generation, while remaining scalable across LLM sizes and embedding dimension $K$. While we acknowledge that more complex or multivariate datasets could lead to longer contexts, our current experiments suggest that the method scales well within practical margins.
>
> ## 3. Efficient generation of multivariate data
> To evaluate whether our structured tabular transformation effectively supports the generation of multivariate time series, we analyse the preservation of inter-variable dependencies in the generated embeddings. Specifically, we compute the Frobenius distance between the correlation matrices of the real and generated embeddings. This distance is normalised to account for differences in embedding scales (k=3,5,7,10) and serves as a proxy for how well the generated data retains the structural relationships observed in the training data.
>
> Table below aggregates result over 5 seeds:
>
> |Dataset|Variables|K = 3*k|Frobenius Distance|
> |-|-|-|-|
> |bikesharing|['cnt','temp','hum']|9|0.434±0.015|
> || |15|0.315±0.011|
> || |21|0.291±0.011|
> || |30|0.256±0.007|
> |||||
> |etth1|['HUFL','MUFL','OT']|9|0.378±0.012|
> || |15|0.321±0.022|
> || |21|0.294±0.010|
> || |30|0.278±0.005|
> |||||
> |traffic|['Junction_1','Junction_2','Junction_3']|9|0.435±0.007|
> || |15|0.351±0.008|
> || |21|0.318±0.004|
> || |30|0.281±0.009|
>
>
> As the embedding dimension increases, the **decreasing Frobenius distance** indicates an enhanced ability of the LLM to capture and reproduce the structured multivariate relationships. This trend suggests that **larger embedding spaces** enable the model to more **effectively learn intricate cross-channel dependencies**, which are essential for faithful multivariate time series generation.
>
> ## Clarification on utility-based evaluation
> Our utility-based metric stems from a key limitation in standard synthetic data evaluation: metrics like distributional similarity or distances often favor models that **closely replicate the training data**, potentially leading to overfitting and reduced generalization. This contradicts the central goal of synthetic data generation, which, in our case, is to produce data that not only resembles the real distribution but also **generalizes well to support downstream tasks**. In this paper, we use **forecasting** as the primary utility benchmark, where synthetic data is employed to fine-tune a predictive model. That said, **expanding utility-based evaluation** to include other real-world task, such as **anomaly detection** or **classification**, presents a promising avenue for future research.
>
> ##  Clarification on LLM fine-tuning
>
> We employ **standard full fine-tuning** of the LLM to adapt it to the structured time series generation task. At this stage, we **do not use PEFT methods** (e.g., LoRA or adapters), as our focus was to demonstrate that **even lightweight models (e.g., GPT-2)** can effectively learn compact tabular embeddings with minimal training data.
>
> **Notably, we achieve strong results without any specialized fine-tuning optimization**, which highlights the accessibility and robustness of the proposed framework. This suggests that SDForger is effective even under default training setups, without requiring task-specific or resource-intensive configurations.
>
> That said, we recognize the **benefits of parameter-efficient fine-tuning**, especially for **scaling** to larger models, supporting **continual learning**, or **domain adaptation**. We consider this a promising direction for future work.

---

> > ### Comment · Reviewer_V1ut · 2025-08-05
> >
> > Thank you for the authors' responses, which addressed most of my concerns. I acknowledge the efficiency of the proposed method and its performance in multivariate generation. While I no longer have any significant doubts, I believe that the true value of this work may be more apparent when applied to larger LLMs. Overall, I will maintain my positive evaluation.

---

### Note · Authors · 2025-08-13

**SDForger** is a LLM-based framework for multivariate time-series generation that bridges structured temporal data and natural language modeling. It combines compact basis embeddings (FastICA or PCA) that capture temporal and inter-variable structure while decoupling from sequence length, with structured text-to-sequence generation via fine-tuned autoregressive LLMs. This enables efficient modeling of long sequences, robust correlation preservation, and adaptability to new domains. Its lightweight, flexible design works with small models and modest training data, while its multivariate and multimodal readiness allows conditioning on categorical or textual covariates for richer, context-aware generation.

In the rebuttal, we addressed reviewers' concerns with additional experiments and clarifications:
- **Embedding dimension choice and efficiency**: We ran experiments, showing clear alignment between dataset complexity and optimal embedding size, and proposed a variance-explained criterion for adaptive selection.
- **Correlation preservation**: Normalized Frobenius distance results confirm that higher embedding dimensions improve structural fidelity in complex datasets without sacrificing simpler ones.
- **Architectural positioning**: We clarified trade-offs with explicit correlation-preserving models, noting SDForger’s advantage in flexibility and potential to integrate new decomposition methods.
- **Data leakage**: We explicitly confirmed that only training splits are used for tuning/training, avoiding leakage.

We will incorporate all clarifications in the camera-ready, add prompt examples in the appendix, and expand the discussion of trade-offs.

Looking forward, SDForger’s modularity opens promising research avenues: exploring **nonlinear** or **multivariate-aware embeddings** (e.g., AEs, Multivariate FPCA), integrating **parameter-efficient** fine-tuning (LoRA, adapters) for scalability, expanding **multimodal conditioning** for richer context integration, and broadening our **utility-based evaluation** to more domains and beyond forecasting improvement.
We believe SDForger provides a simple yet powerful new baseline for multivariate time-series generation, offering both practical utility and a platform for future research in **multimodal time-series synthesis**.

---

### Decision · Program_Chairs · 2025-09-17

**Decision:**

Accept (poster)

**Comment:**

This paper studies time series generation using large language models. The submission received borderline scores (3 borderline accepts and 1 accept). Reviewers noted the clear writing and well-structured presentation, and the evaluation was significantly improved after the rebuttal.

However, concerns remain. Reviewer V1ut questioned the utilization of large language models; Reviewer Ck4d highlighted the limited evaluation datasets and weak empirical evidence; Reviewer 2HPq noted the limited technical novelty, largely combining established methods such as ICA and LLM fine-tuning; and Reviewer Exqs pointed out missing implementation details such as prompt examples.

I have carefully read the paper and the reviewer–author discussions, and I find these limitations remain valid. Considering the competitive nature of this year’s submissions, **I lean toward recommending acceptance, though I would not object if this recommendation were downgraded**.

I strongly encourage the authors to substantially revise the paper in line with the reviewers’ comments, particularly by strengthening the motivation and empirical comparisons with alternative embedding methods, providing deeper analysis on the role of large language models, and adding missing implementation details.